# Learning and Aligning Single-Neuron Invariance Manifolds in Visual Cortex

**Mohammad Bashiri**[1,2,*,†], **Luca Baroni**[3,*], **Ján Antolík**[3], **Fabian H. Sinz**[2,4]
[1]Noselab GmbH, Munich, Germany
[2]Department of Computer Science, University of Göttingen, Germany
[3]Faculty of Mathematics and Physics, Charles University, Prague, Czechia
[4]Campus Institute Data Science (CIDAS), University of Göttingen, Germany
[*]Equal contribution, [†]mohammadbashiri93@gmail.com

## Abstract

Understanding how sensory neurons exhibit selectivity to certain features and invariance to others is central to uncovering the computational principles underlying robustness and generalization in visual perception. Most existing methods for characterizing selectivity and invariance identify single or finite discrete sets of stimuli. Since these are only isolated measurements from an underlying continuous manifold, characterizing invariance properties accurately and comparing them across neurons with varying receptive field size, position, and orientation, becomes challenging. Consequently, a systematic analysis of invariance types at the population level remains under-explored. Building on recent advances in learning continuous invariance manifolds, we introduce a novel method to accurately identify and align invariance manifolds of visual sensory neurons, overcoming these challenges. Our approach first learns the continuous invariance manifold of stimuli that maximally excite a neuron modeled by a response-predicting deep neural network. It then learns an affine transformation on the pixel coordinates such that the same manifold activates another neuron as strongly as possible, effectively aligning their invariance manifolds spatially. This alignment provides a principled way to quantify and compare neuronal invariances irrespective of receptive field differences. Using simulated neurons, we demonstrate that our method accurately learns and aligns known invariance manifolds, robustly identifying functional clusters. When applied to macaque V1 neurons, it reveals functional clusters of neurons, including simple and complex cells. Overall, our method enables systematic, quantitative exploration of the neural invariance landscape, to gain new insights into the functional properties of visual sensory neurons.

## 1 Introduction

To understand visual sensory processing, it is crucial to examine not only the coding properties of individual neurons but also how these coding properties are organized across the entire neural population. Two fundamental principles of sensory coding are selectivity and invariance: visual neurons exhibit selectivity to specific features such as orientation, while maintaining invariance to other transformations such as translation (Hubel & Wiesel, 1962; Riesenhuber & Poggio, 1999). For example, in the primary visual cortex (V1), simple cells are selective to the precise phase of a stimulus, whereas complex cells exhibit phase invariance, responding consistently regardless of phase shifts (Hubel & Wiesel, 1962). These computational properties enable neurons to maintain informative yet stable responses across diverse inputs, thus supporting reliable object recognition and visual processing.

Recently, data-driven approaches using Deep Neural Networks (DNNs), acting as *digital twins* of sensory neurons, have dramatically advanced our ability to model neural responses (Yamins & DiCarlo, 2016; Cadena et al., 2019; Lurz et al., 2020; Klindt et al., 2017). These approaches enable the identification of a single maximally exciting input (MEI) (Walker et al., 2019; Bashivan et al., 2019) or a diverse set of nearly maximally exciting stimuli that outline the invariance space of the

neuron (Ding et al., 2023; Cadena et al., 2018). In this context, an invariance manifold can be defined as the set of stimuli that elicit near-maximal responses from a neuron. These stimuli are related by transformations to which the neuron is invariant, meaning that the neuron's response remains constant (or nearly so) across these variations. For instance, the invariance manifold of a complex cell exhibiting phase invariance is the set of Gabor patterns that differ only in phase and maximally excite it. While invariances can also be explored at suboptimal response levels, focusing on peak response levels highlights the stimuli to which the neuron is most selective. Most existing methods, however, do not readily enable analyses to compare the invariance manifolds of neurons across populations and to identify clusters of neurons sharing common invariances. Ideally, such a method should meet the following requirements:

- It should accurately capture the invariance manifold of individual neurons, ensuring that subsequent analyses faithfully reflect neural coding.

- It should enable the identification of common invariance properties while disregarding differences in selected 'nuisance' receptive field (RF) attributes such as position, size, and orientation.

- It should allow for a meaningful measure of the similarity of invariance properties across neurons.

Current approaches do not satisfy all of these requirements simultaneously.

In this paper, we propose a novel approach for aligning and comparing invariance manifolds across a population of neurons, fulfilling the above-mentioned requirements and allowing clustering of neurons based on their invariance properties. Our method first identifies the invariance manifold of individual neurons (Fig. 1 top panel) using the same approach as Baroni et al. (2023). It then learns an affine transformation on the pixel coordinates such that the same manifold activates another neuron as strongly as possible, effectively aligning their invariance manifolds spatially (Fig. 1 middle panel). Neurons whose invariance manifolds strongly activate each other likely share similar computational roles (Fig. 1 bottom panel). By performing this pairwise alignment across a population of neurons, our method systematically uncovers shared invariance properties, irrespective of differences in their receptive field attributes, allowing for clustering of neurons based on their invariances.

We tested our method on simulated neurons and macaque V1 neurons modeled by a response-predicting DNN. The simulated neurons were carefully designed to display a range of invariance properties, and our method successfully learned and aligned their corresponding invariance manifolds, enabling accurate identification of the ground truth functional types. When applied to a response-predicting model of macaque V1 neurons, our approach captured and aligned the invariance manifolds of these neurons, uncovering multiple functional clusters, including, but not limited to, the expected clusters of canonical simple and complex cells.

Overall, our method generalizes effectively from synthetic to macaque V1 neurons, offering a robust framework for systematically analyzing invariance properties across sensory neurons, to deepen our understanding of the computational roles of individual neurons and gain new insights into the broader organizational principles governing neural populations.

## 2 RELATED WORK

Our method is uniquely situated at the intersection of characterizing neuronal invariances and clustering neurons based on shared functional properties. The works discussed below are closely related to one or more aspects of our approach, either addressing neuronal invariances or functional clustering of neurons.

**Neuronal invariances** Recent pioneering work demonstrated the use of deep learning to characterize neural selectivity by identifying maximally exciting inputs (MEIs), which were subsequently validated through *in-vivo* experiments (Bashivan et al., 2019; Walker et al., 2019; Ponce et al., 2019). Using response-predicting DNNs, Cadena et al. (2018) and Ding et al. (2023) characterized neural invariances by generating multiple such MEIs for a neuron by directly optimizing the pixel values while encouraging diversity across the MEIs using an additional regularization term.

**Limitations of discrete approximations** While these methods marked a step forward in characterizing invariances, they approximate the underlying continuous invariance manifold through a finite set

of points. This discretized approach risks leaving gaps in the manifold, potentially resulting in a less precise representation of the invariance, or requires a large number of samples, making it computationally challenging to capture its full complexity. This could result in challenges for single-neuron invariances that involve more complex transformations along the underlying manifold, as simply interpolating between discrete points may not reliably produce new samples on the manifold.

**Learning continuous invariance manifolds**  Recently, Baroni et al. (2023) proposed an alternative approach that combines contrastive learning with implicit neural representations (INRs) to map a continuous latent input onto the invariance manifold. This method captures the manifold as a continuous entity, addressing the discretization issue from previous methods. Similarly, Wang & Ponce (2022a;b) introduced a framework for modeling neural invariances at different activation levels, providing theoretical insights into their geometry and examining variations in tuning landscapes across the ventral stream. In contrast to Baroni et al. (2023), which learned continuous invariance manifolds using INRs, Wang & Ponce (2022a;b) used a pretrained generative model to explore invariances in its latent space, capturing diverse patterns within the naturalistic image space learned by the generative model. Although these methods advance our understanding of invariance manifolds, they do not provide a mechanism to compare invariances across neurons at the population level, a gap we aim to address in this study.

**Functional clustering of neurons**  Reliable comparison of neurons based on encoding properties requires separating shared nonlinear computations from variations in nuisance RF properties, such as location, size, and orientation. For example, two simple cells with the same RF pattern but different locations and orientations should be grouped in the same cluster. While multiple methods have been developed for clustering neurons based on their encoding properties, they typically do not fully isolate the nonlinear computations from the nuisance RF properties or use clustering metrics that are often difficult to interpret.

**Model-based clustering**  Ustyuzhaninov et al. (2019) employed a model-based approach to cluster neurons based on feature weights in the readout layer of a rotationally-equivariant response-predicting model. This method effectively clusters neurons with similar encoding properties, regardless of differences in the orientation and position of their RFs. However, this approach does not account for RF size variations, a property known to vary with eccentricity in higher mammals' visual systems (Wilson & Sherman, 1976; Cavanaugh et al., 2002; Harvey & Dumoulin, 2011). Additionally, this approach risks incorrectly separating neurons with similar encoding properties, as the same neuronal response function can be realized by different combinations of features. Finally, a metric on feature weights does not readily translate into an interpretable metric of neuronal similarity.

**Stimulus-based clustering**  Burg et al. (2024) introduced a technique that clusters neurons by identifying the most discriminative stimuli across groups of neurons, but does not account for variations in RF orientation or size. Similarly, Willeke et al. (2023) and Tong et al. (2023) employed contrastive learning to cluster neurons with similar maximally exciting inputs (MEIs) in macaque V4 and mouse visual cortex, respectively. However, their method relies on dimensionality reduction of single-neuron MEIs, which may fail to cluster neurons exhibiting invariance properties reliably. For example, two images from the invariance manifold of the same neuron could end up in different clusters, leading to inconsistent cluster assignment depending on the image used.

## 3  METHOD

Our approach consists of two stages: *Template Learning* and *Template Matching*. The first stage identifies the continuous invariance manifold of a single neuron, and the second stage aligns this learned manifold with another neuron to maximize its activity, evaluating their similarity based on shared invariance properties (Fig. 1).

### 3.1  TEMPLATE LEARNING

**Implicit Neural Representation of the invariance manifold**  In the template learning step, following Baroni et al. (2023), we use Implicit Neural Representations (INRs) to parameterize the continuous invariance manifold of a neuron modeled by a response-predicting DNN, generating diverse images that elicit maximal activity. INRs are neural networks that represent continuous signals, such as images or 3D shapes, by mapping coordinates to signal values without relying on discrete grid-based

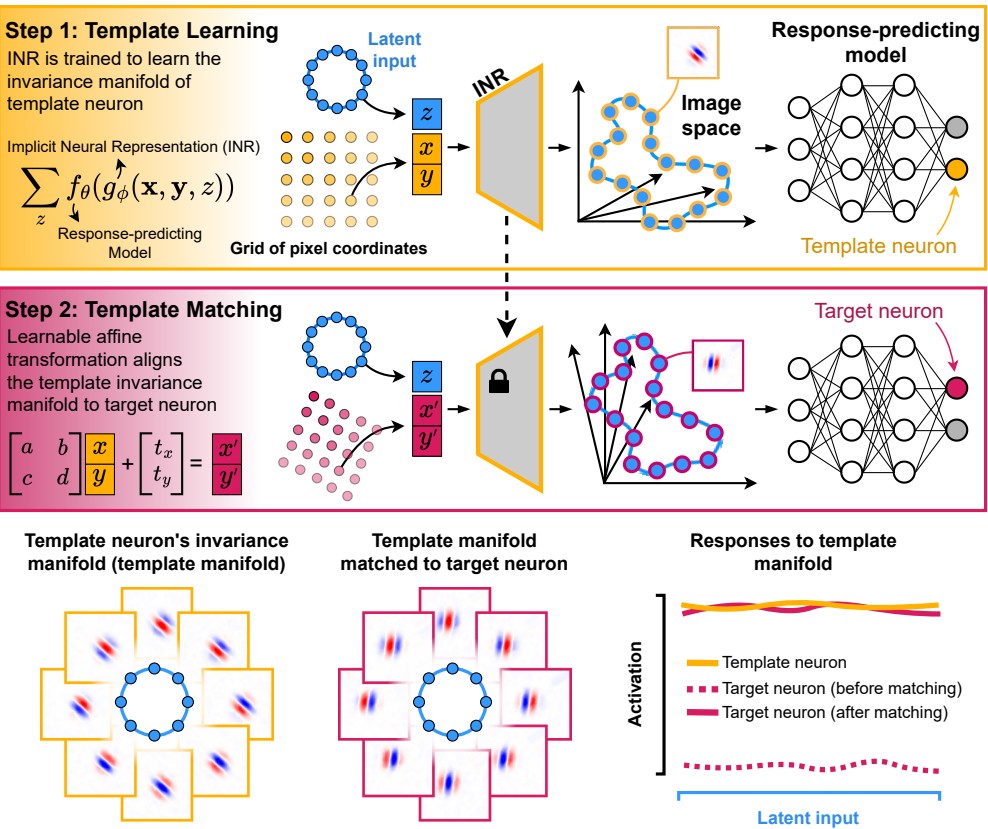

Figure 1: Overview of the method. **Top panel)** Template learning. An Implicit Neural Representation (INR), mapping from pixel coordinates and a latent space, generates a manifold of images that maximally, and equally, excite a neuron (referred to as the template neuron). **Middle panel)** Template matching. The same INR is used to generate the invariance manifold of the template neuron. An affine transformation on the pixel coordinates $(x, y)$ is learned to align the template invariance manifold to a target neuron such that it activates the target neuron as strongly as possible. **Bottom panel)** Left: Template neuron's invariance manifold (referred to as the template manifold). Center: Template manifold after the template matching step, *i.e.*, aligning to target neuron. Right: Schematic representation of the neurons' responses to different images of the template manifold. The template manifold maximally activates the template neuron but does not activate the target neuron. However, after matching, if the template neuron and target neuron present similar invariance properties, it maximally activates the target neuron.

representations (Genova et al., 2019; Atzmon & Lipman, 2020; Sitzmann et al., 2019; Mildenhall et al., 2021). In the context of image generation, an INR maps 2D pixel coordinates $(x, y)$ to pixel values $g_\phi(x, y)$, producing a continuous, resolution-independent image from the input grid of pixel coordinates. Similar to Baroni et al. (2023), we introduce a periodic 1D latent variable $z \in [0, 2\pi)$ as an additional input to the INR, enabling it to generate multiple images from the same pixel grid and produce a family of images $\{g_\phi(\mathbf{x}, \mathbf{y}, z)\}_{z \in \mathcal{Z}}$ by varying $z$. For brevity, we might represent $g_\phi(\mathbf{x}, \mathbf{y}, z)$ as $g(z)$ in the following equations, omitting the dependence on $\mathbf{x}, \mathbf{y}$, and the parameters $\phi$, which remain implicit.

While we generally follow the approach of Baroni et al. (2023), we introduced several modifications, including changes to the architecture and the use of positional encoding. Specifically, we implemented the INR as a fully connected neural network with 4 hidden layers, each containing 50 hidden units with a Tanh nonlinearity. Since we aim to generate grayscale images, the output layer consists of a single unit with a Tanh activation, producing values between -1 and 1. Additionally, we used positional encoding with random Fourier features (Tancik et al., 2020), as used by Baroni et al. (2023), on the pixel coordinates $(x, y)$ and extended it to the latent variable $z$. When applied to the

pixel coordinates, this encoding enables control over the spatial frequency of the generated images, whereas its application to the latent variable allows the INR to capture more complex variations along the manifold, even with small changes in $z$.

**Objective function** The INR parameters $\phi$ are trained to generate a diverse set of images that collectively span the invariance manifold of a target neuron, ensuring all images elicit maximal activation. The images generated by the INR are fed to a response-predicting DNN $f_\theta(g_\phi(\mathbf{x}, \mathbf{y}, z_i))$ with fixed parameters $\theta$, which evaluates the activation of the target neuron in response to the generated images (Fig. 1 top panel). Before being fed to the DNN, the generated images are linearly scaled to satisfy a specific norm constraint. Combined with the activity maximization objective, this ensures that the pixel variability in the generated images represents only features relevant to the neuron and prevents saturation of pixel values. To ensure diversity of patterns across the generated images, spanning the underlying invariance manifold, we use a contrastive objective (Chen et al., 2020) that encourages variations across the images. Combined with the objective of maximizing the neuron's response, the complete objective function (to maximize) for template learning is:

$$\mathcal{L}_{TL} = \frac{1}{N} \sum_{i=1}^{N} \left( \frac{\alpha_i}{\alpha_{\text{MEI}}} + \lambda \cdot \log \left( \frac{\frac{1}{N_+} \sum_{\mathbf{z}_j \in \mathcal{Z}_+^i} \exp\left(\text{sim}(g(\mathbf{z}_i), g(\mathbf{z}_j))/\tau\right)}{\frac{1}{N_-} \sum_{\mathbf{z}_k \in \mathcal{Z}_-^i} \exp\left(\text{sim}(g(\mathbf{z}_i), g(\mathbf{z}_k))/\tau\right)} \right) \right), \quad (1)$$

where $N$ is the total number of generated images corresponding to the number of latent points $z$. $\frac{\alpha_i}{\alpha_{\text{MEI}}}$ is the activation $\alpha_i = f_\theta(g_\phi(\mathbf{x}, \mathbf{y}, z_i))$ of the target neuron, normalized by its response to its MEI, obtained by standard pixel optimization (Appendix A). Although different latent points were chosen to be equally spaced on a grid, the grid was jittered at each training step to continuously cover the full range of the latent input. The strength of the contrastive objective is controlled by $\lambda$. The contrastive term uses cosine similarity $\text{sim}(g(z_i), g(z_j))$ between images generated from latent points $z_i$ and $z_j$, scaled by temperature $\tau$. Positive examples $\mathcal{Z}_+^i$ come from latent points near $z_i$, while negative examples $\mathcal{Z}_-^i$ come from distant points, with $N_+$ and $N_-$ being their respective counts. This formulation encourages similarity between images corresponding to nearby latent points while making images corresponding to distant latent points different from each other, promoting a smooth and diverse learned manifold. Additional details on the template learning step can be found in Appendix B, and an analysis of the method's robustness to varying definitions of "near" and "distant" latent points is provided in Appendix C.

## 3.2 TEMPLATE MATCHING

**Aligning the learned invariance manifold** After the template learning step, where we identify the invariance manifold of the template neuron, the second stage focuses on aligning this template manifold with another neuron (referred to as the *target neuron*). We refer to this step as Template Matching (Fig. 1 middle panel). In this step, the template manifold is fixed by freezing the INR parameters $\phi = \phi^*$. To spatially transform the template, we learn an affine transformation $A_\psi$ applied to the pixel coordinates $(x, y)$ such that images sampled from the template maximize the response of the target neuron $\tilde{f}_\theta(g_{\phi^*}(A_\psi(\mathbf{x}, \mathbf{y}), z))$.

**Objective function** The objective function for optimizing the parameters $\psi$ of the affine transformation matrix is similar to Eq. 1 but without the contrastive learning term since the INR parameters are not being optimized in this step and the invariance manifold is fixed:

$$\mathcal{L}_{TM} = \frac{1}{N} \sum_{i=1}^{N} \frac{\tilde{\alpha}_i}{\tilde{\alpha}_{\text{MEI}}}, \quad \tilde{\alpha}_i = \tilde{f}_\theta(g_{\phi^*}(A_\psi(\mathbf{x}, \mathbf{y}), z_i)), \quad (2)$$

where $N$ is the total number of images generated from the manifold, $\tilde{\alpha}_i$ is the activation of the target neuron when presented with the transformed image $g_{\phi^*}(A_\psi(\mathbf{x}, \mathbf{y}), z)$, and $\tilde{\alpha}_{\text{MEI}}$ is the response of the target neuron to its MEI. By learning the affine transformation, we align the spatial structure of the template neuron's manifold to that of the target neuron. This alignment allows us to probe the degree of shared invariance between the two neurons, based on how well the transformed manifold activates the target neuron (Fig. 1 bottom panel). The activation metric used for alignment is immediately interpretable because it tells us how much the template manifold can drive another neuron. Additional details on the template matching step can be found in Appendix D.

Importantly, while we focus on affine transformations, our formulation can be extended to nonlinear (invertible) distortions of space or subsets of affine transformations, allowing control over how flexible or constrained the invariance alignment should be. Additionally, the continuous nature of INRs prevents artifacts typically introduced by discrete pixel coordinate transformations.

### 3.3 CLUSTERING BASED ON INVARIANCE ALIGNMENT

By applying the template learning and template matching steps to all pairs of neurons in the population, we construct an activation matrix, where each row corresponds to one neuron (the template neuron) and each column to another neuron (the target neuron). Each entry $(i, j)$ in this matrix represents the average response of neuron $j$ to the images of the invariance manifold of neuron $i$ after matching. To make sure that activations are comparable across neurons despite differences in their intrinsic excitability, we used relative activations as opposed to absolute responses. The relative response for any given neuron was computed as: $(\alpha - \alpha_0)/(\alpha_{\text{MEI}} - \alpha_0)$ where $\alpha$ is the absolute response, $\alpha_0$ is the baseline response (*i.e.*, response to a gray image), and $\alpha_{\text{MEI}}$ is the MEI response. This yields relative activation values where $0$ corresponds to the baseline response and $1$ to the MEI response.

The activation matrix reflects functional relationships between neurons by capturing how their invariance manifolds activate each other. Importantly, because neurons exhibit nonlinear response characteristics, clustering based on the activation matrix offers a better understanding of functional similarity than clustering directly using invariance manifolds or MEIs, as distances in image space do not necessarily have a one-to-one correspondence with functional differences between neurons.

Additionally, notice that the matrix is not symmetric. Specifically, if $\mathcal{M}_1$ and $\mathcal{M}_2$ are the invariance manifolds of the template and target neuron, respectively, a high activation means that $\mathcal{M}_1 \subseteq \mathcal{M}_2$, but not necessarily vice versa. For example, if we consider a simple and a complex cell, after the alignment of their invariance manifold, the Gabor pattern of a simple cell would maximally activate a complex cell, while the invariance manifold of a complex cell would not maximally activate a simple cell. Therefore, this asymmetry reflects a meaningful difference between neurons' computational roles and the relationship between their invariance manifolds.

Once the activation matrix is constructed, we cluster neurons by applying a standard agglomerative hierarchical clustering on the rows of the activation matrix. Each row corresponds to a single neuron and represents how strongly its invariance manifold activates other neurons after matching. We use cosine dissimilarity as the distance metric for clustering, which helps focus on the relative activation patterns between neurons rather than the magnitude of the activation, ensuring that functional relationships are properly captured.

## 4 EXPERIMENTS AND RESULTS

### 4.1 VALIDATION ON SIMULATED NEURONS

As a first step, we validated our approach using a set of Gabor-based simulated neurons to verify and quantify its ability to accurately capture and align neural invariances across different neuron types. The main advantage of this simulated environment is that we have access to the ground truth invariances for each neuron type, enabling precise benchmarking of the accuracy of the method.

The simulated neurons consisted of several types, including simple and complex cells, as well as additional neurons exhibiting non-trivial, arbitrary invariance properties (Fig.2A, top row). Simple cells were modeled as linear-nonlinear systems using a single Gabor filter and complex cells were modeled using a set of Gabor filters with shared parameters but varying phases, achieving phase invariance by selecting the highest response across filters. Neurons with more complex invariances were similarly modeled using filters obtained as combinations of Gabors with varying arbitrary parameters and selecting the highest response across filters to achieve the desired invariance. For each neuron type, we created six instances, resulting in a total of 36 neurons. Neurons of the same type differed from one another by an affine transformation of their receptive fields, including variations in position, orientation, scale, and shear (Fig.2A, bottom row). To standardize the response of the neurons and facilitate the comparison of results, we normalized the Gabor filters to have a maximum activation of 1 in response to stimuli of unitary norm. Further details on the simulation can be found in Appendix E.

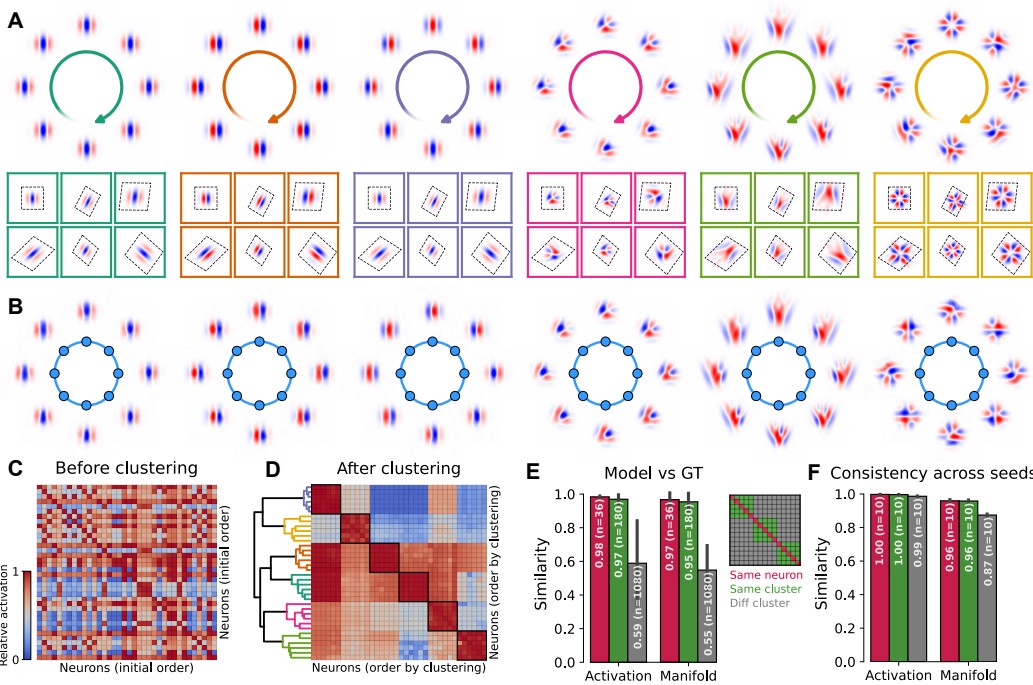

Figure 2: Results on simulated neurons. **A)** Types of Gabor-based simulated neurons, each associated with a distinct color. Top row: invariance manifold of a representative neuron for each type. Left to right: even simple cells, odd simple cells, complex cells, and three types of neurons with selectivity and invariance to more complex patterns. Note that images for simple cells are identical, indicating their lack of invariance. Bottom row: For each neuron type we model 6 neurons that differ by an arbitrary affine transformation illustrated by dotted lines around each neuron's receptive field. **B)** Learned invariance manifolds for representative neurons shown in Panel A top row. **C)** Activation matrix resulting from the pairwise alignment of neural invariances. Each row represents how the invariance manifold of one neuron activates others post-alignment, while each column shows how a specific neuron is activated by the manifolds of other neurons. **D)** Activation matrix ordered by applying a hierarchical clustering algorithm. The block diagonal pattern corresponds to identified clusters reflecting different neuron types. Dendrogram colors correspond to neuron types in Panel A. **E)** Comparison of predictions vs. ground truth. Left barplot: activations from learned manifolds (red), aligned manifolds of neurons within the same cluster (green), and aligned manifolds of neurons from different clusters (gray). A similarity of 1 means the invariance manifold activates the neuron as high as its MEI. Right barplot: similarity scores between the ground truth and the learned manifolds (red), aligned manifold of neurons within the same cluster (green), and aligned manifold of neurons from different clusters (gray). **F)** Consistency of results across 5 seeds, *i.e.*, $n = 10$ unique pairwise combinations. Left barplot: similarity scores of activation matrices across seeds. Similarity of diagonal entries is shown in red, off-diagonal same cluster is shown in green, and off-diagonal different cluster is shown in gray. Right barplot: similarity scores of learned manifolds (red), aligned manifolds of neurons within the same cluster (green) and from different clusters (gray), across seeds.

We applied the template learning and template matching steps to all pairs of simulated neurons resulting in an activation matrix where row $i$ shows how strongly the invariance manifold of neuron $i$ activated all other neurons (Fig. 2C). While the neurons' order is randomized in this matrix, the high activation in the diagonal entries shows that our method successfully recovered the known invariance transformations for each neuron, generating invariance manifolds that activated the neurons close to their maximal activation (see Fig. 2B for examples of the learned invariance manifold). We then applied hierarchical clustering, using cosine dissimilarity as a distance metric, to the rows of the activation matrix, accurately recovering the predefined neuron types (Fig. 2D) and showing that our method successfully aligned the invariance manifolds of simulated neurons with similar type, despite differences in their nuisance RF properties.

We further quantified deviations from the ground truth by analyzing the learned and aligned manifolds as well as the corresponding activations. For a given neuron, we computed the activation similarity between the ground truth $act_{GT}$ and the learned manifold activation $act$ as $1 - |act_{GT} - act|$. Similarly, we computed manifold similarity by measuring how well the images on one manifold match the closest images on the other manifold, averaged bidirectionally. This captures the degree to which the two manifolds overlap in terms of their most similar images (for more details refer to Appendix F). Our method effectively yielded activation values near the maximum not only in response to the learned invariance manifold of each neuron but also in responses to invariance manifolds of other neurons within the same type, after being matched (Fig. 2E left, red and green bar). Notably, when this alignment was performed between neurons of different types, the resulting activation was consistently lower, as expected (Fig. 2E left, gray bar). When comparing the learned and aligned manifolds with ground truth manifolds, we observed high similarity scores for manifolds aligned within the same cluster, indicating that invariances were successfully aligned (Fig. 2E right, red and green bar). Alignments between different clusters yielded lower similarity scores, highlighting differences in invariance properties across distinct neuron types (Fig. 2E right, gray bar).

Additionally, we assessed the consistency of our method across different initializations (seeds), at both the activation and manifold levels. For comparing the activations across seeds, we defined an activation similarity metric as $1 - |act_i - act_j|$, which measures the closeness of activations between seed $i$ and $j$, averaged across entries in the activation matrix. Similarly, we computed a similarity score of the corresponding learned and aligned manifolds across seeds to evaluate consistency in manifold learning and alignment, directly using images sampled from the manifolds. As depicted in Fig. 2F, our analyses show that our results on the synthetic data are robust, with high consistency observed both at the level of activation and in the alignment of invariance manifolds.

Finally, in Appendix G, we present a detailed comparison of our approach with two model-based clustering methods introduced by Klindt et al. (2017) and Ustyuzhaninov et al. (2019). Specifically, we show that while these methods effectively account for shifts (Klindt et al., 2017) or both shifts and rotations (Ustyuzhaninov et al., 2019), they do not extend to the broader class of affine transformations. Notably, the comparison also highlights that our method is model-agnostic, meaning that it can be applied to any robust response-predicting model of biological neurons, further demonstrating its flexibility.

## 4.2 APPLICATION ON MACAQUE V1 NEURONS

We then applied our method to a response-predicting model of a population of biological neurons recorded from the primary visual cortex of two macaques, using the response-predicting model from Baroni et al. (2023) trained on the publicly available dataset from Cadena et al. (2024). Briefly, this dataset contains neural responses from 458 V1 neurons recorded across 32 sessions as the macaques viewed up to 15,000 unique grayscale ImageNet images per session, while fixating. Images were shown for 120ms, and neuronal responses were measured by counting spikes from 40 to 160ms post-image onset. The model from Baroni et al. (2023) is an ensemble of three DNN-based models which were trained to predict these neural responses, approaching state-of-the-art predictive performance with a correlation of 0.73 between predicted and observed trial-averaged responses. We selected the 100 best-predicted neurons (53 neurons from Macaque 1 and 47 from Macaque 2) to ensure accurate modeling of their encoding properties (Fig. S3).

Applying the template learning step, our method learned invariance manifolds (Fig. 3A) that maximally activate the neurons (Fig. 3B, diagonal entries). We then performed the template matching step to align the invariance manifolds across neurons, resulting in a 100-by-100 activation matrix. As with the simulated data, we applied hierarchical clustering to this matrix to group neurons based on how their invariance manifold activated other neurons after matching (Fig. 3B).

To identify meaningful clusters, we selected a dendrogram cut-off threshold that minimized the number of clusters while ensuring that neurons within each cluster shared similar invariance properties, as verified through visual inspection of their invariance manifolds. This resulted in five clusters containing multiple neurons (Fig. 3A). We identified two large clusters corresponding to simple and complex cells (Fig. 3 A-B, Clusters 1 and 2). Neurons in Cluster 1 displayed selectivity to Gabor patterns with no invariances, whereas neurons in Cluster 2 exhibited phase invariance. The remaining smaller clusters did not show uniform orientation tuning across their receptive fields, but

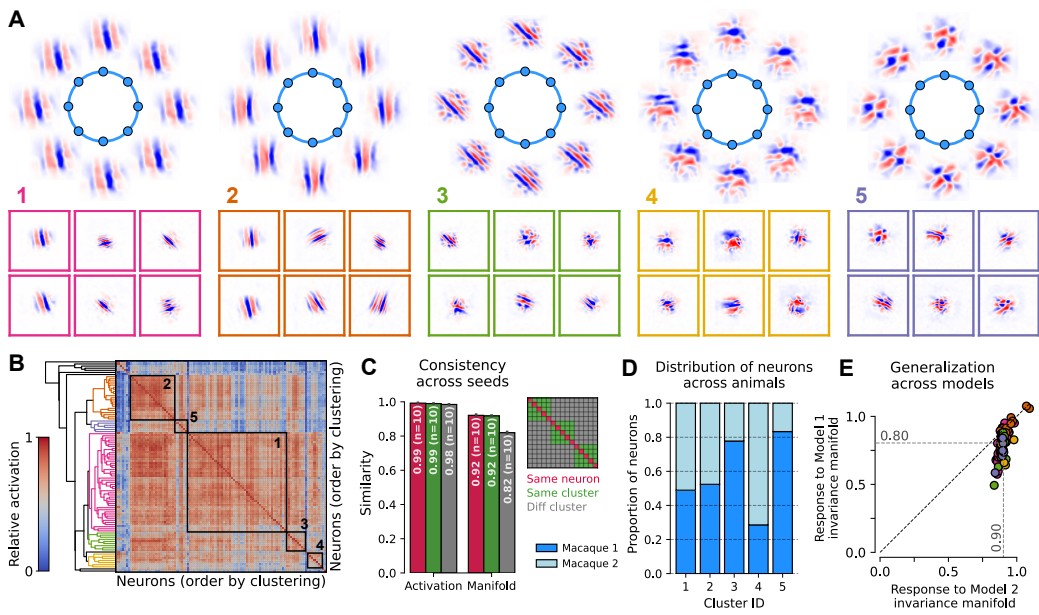

Figure 3: Results on macaque V1 neurons. **A)** Examples of learned invariance manifolds, organized by clusters identified in panel B. Each cluster is assigned a numerical ID and color. The top row shows invariance transformations (or their absence) for representative neurons from different clusters. The bottom row displays an example image within the invariance manifold for multiple neurons for each cluster. **B)** Activation matrix ordered by hierarchical clustering applied to rows, revealing clusters of neurons with shared invariance properties. **C)** Consistency of results across 5 seeds. Left barplot: similarity scores of activation matrices across seeds. Similarity of diagonal entries is shown in red, off-diagonal same cluster shown in green, and off-diagonal different cluster shown in gray. Right barplot: similarity scores of learned manifolds (red), aligned manifolds of neurons within same cluster (green) and from different clusters (gray), across seeds. **D)** Distribution of neurons within each cluster across different animals. **E)** Generalization across models. Invariance manifolds from model 1 (which was used for the previous analyses) were shown to model 2 (a model with different architecture predicting the same neurons). The y-axis shows the response of model 2 to model 1's invariance manifolds, and the x-axis shows the response of model 2 to its own invariance manifolds.

instead exhibited selectivity to more complex patterns, diverging from the traditional classification of V1 neurons into simple and complex cells. For example, Cluster 5 contained neurons selective to checkerboard-like texture patterns and showed invariance to local alterations of these patterns. All identified clusters contained neurons recorded from both macaques (Fig. 3D), confirming that our analysis did not detect inter-subject differences, reflecting consistent sensory coding across the two animals. As we did with the simulated neurons, we tested the robustness of our method applied to the macaque V1 neurons across different seeds. Our method produced consistent results in terms of the learned and aligned manifolds (Fig. 3C) and cluster consistency (Adjusted Rand Index = 0.75).

Finally, to rule out the possibility that our results were due to idiosyncrasies of the response-predicting model, we trained a second model with a different architecture (details in Appendix H) on the same set of neurons (Fig. S3B). We refer to the initial model as Model 1 and the newly introduced model as Model 2. We presented the invariance manifolds obtained from Model 1 to their corresponding neurons in Model 2. Then, we learned the invariance manifolds of the same neurons directly from Model 2 and compared the activations resulting from these two procedures. On average, the invariance manifold from Model 1 activated the corresponding neuron in Model 2 at 89% of the strength of its own manifold, confirming that the invariance manifolds from Model 1 strongly activate the same neurons in Model 2 and generalize well (Fig. 3E).

Together, these results demonstrate that our method can robustly identify common invariance properties of visual coding in biological neurons.

## 5 DISCUSSION

We introduced a novel method to identify and align the continuous invariance manifolds of visual sensory neurons, enabling systematic characterization of visual coding across neural populations. By combining DNN-based response-predicting models of biological neurons, an INR, and affine transformations to learn and align single-neuron invariance manifolds, we provide a principled approach to quantify and compare invariances irrespective of nuisance receptive field differences. Analyzing the activation matrix resulting from invariance alignment provides insights into the functional differences among neurons, enabling grouping into distinct functional clusters based on shared invariance properties.

When applied to Gabor-based simulated neurons, our method accurately learned the true underlying invariance manifolds and successfully aligned them across neurons, identifying clusters corresponding to the ground truth neuron types. Applying our method to 100 well-predicted macaque V1 neurons, we identified five clusters: two corresponding to traditional simple and complex cells, and smaller clusters with invariances to more intricate patterns. For example, one class responded to checkerboard-like textures and was invariant to phase variations in multiple directions, exhibiting complexity beyond the simple/complex dichotomy.

We currently cannot fully exclude the possibility that these clusters result from recording artifacts or data preprocessing steps, such as spike sorting. Ultimately, dedicated closed-loop neurophysiological experiments would need to be conducted to empirically validate the clusters. However, there is some evidence that these clusters could reflect biological reality. First, all selected neurons exhibited high predictive performance on a test set of images that were not used for training the response-predicting model, suggesting that the model accurately captured the encoding properties of these neurons. Second, all clusters contained cells from both monkeys, indicating that the RF patterns and invariances generalize across animals. Third, the invariance manifolds learned by one model highly activated the same neurons of another model with a different architecture, indicating that the clusters, which are based on the learned invariances, are likely not an artifact of the network used to fit the cells, but reflect a property of the underlying data. Thus, our method could have potentially found novel types of invariances in macaque V1 neurons, similar to the more complex invariances found in mouse visual cortex (Ding et al., 2023).

While our approach has merits, it is not without limitations. First, the computational complexity of the method scales quadratically with the number of neurons, making it relatively expensive to apply on larger neural populations. Second, aligning complex invariance manifolds spatially is not necessarily a convex problem, which could complicate the alignment process for neurons with more intricate invariances. Finally, in this work we focused on one-dimensional invariance manifolds. While this appears to be sufficient for macaque V1 data, other cortical areas might require higher-dimensional invariance manifolds. Nonetheless, Baroni et al. (2023) showed that INRs can be used to learn higher-dimensional invariances.

Looking ahead, several promising directions can be explored to extend this work. Improving the scalability of the approach facilitates its application to larger neural populations. The flexibility of our framework allows exploring transformations beyond affine, enabling alignment of invariances up to more complex transformations. Additionally, our method can be easily applied to other species, potentially identifying novel invariance properties and enabling cross-species comparisons to reveal conserved or divergent functional principles across different neural systems.

In summary, the method we presented is a novel tool for exploring neural invariances in biological data at the population level, enabling the identification of neuron clusters with shared invariance properties, with the potential to uncover new principles of neural computations in the visual cortex. These principles could also inform the design of artificial neural networks, helping to bridge the discrepancies between biological and artificial systems highlighted by Feather et al. (2019; 2023).

REPRODUCIBILITY STATEMENT

The complete implementation of our method, including a Docker container for easy setup, can be found in `https://github.com/sinzlab/laminr`. All relevant training configurations, hyperparameters, and implementation details are provided in the appendix for full transparency. The experiments were conducted using Python 3.9, PyTorch 1.13.1, and CUDA 11.7. However, our method is also compatible with the latest stable versions: PyTorch 2.6.0 and CUDA 12.4. Running the complete pipeline on simulated neurons took approximately 2 hours on a single V100 GPU (one seed, total number of 36 simulated neurons). For macaque V1 neurons (total of 100 neurons), each neuron's template learning and matching experiment was run separately and took approximately 15-20 minutes on a single A100 GPU (one seed).

ACKNOWLEDGMENTS

We would like to thank Noselab GmbH, in particular Thomas Heydler and Arno Schäpe, for supporting MB. LB and JA were supported by the EU Horizon 2020 Maria SklodowskaCurie grant agreement No 861423. JA was supported by the ERDF-Project Brain dynamics, No. CZ.02.01.01\00\22_008\0004643 and by the grant no. 25-18031S of the Czech Science Foundation (GAČR). FHS was supported by the German Research Foundation (DFG): SFB 1233, Robust Vision: Inference Principles and Neural Mechanisms Project-ID 276693517, and SFB 1456, Mathematics of Experiment – Project-ID 432680300. Computing time was made available on the high-performance computers HLRN-IV at GWDG at the NHR Center NHR@Göttingen. The center is jointly supported by the Federal Ministry of Education and Research and the state governments participating in the NHR (www.nhr-verein.de/unsere-partner).

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

# APPENDIX

## TABLE OF CONTENTS

# A  MEI GENERATION

For a given neuron, We obtained the Maximum Exciting Image (MEI) by directly optimizing image pixel values to maximize its response over 1000 steps with a step size of 10. Typically, the optimization converged well before reaching the maximum steps.

Generating single MEIs prior to learning invariance manifolds is critical for two reasons. First, it provides the maximal neuron response $\alpha_{MEI}$, which is used to normalize the balance between the activation term and the contrastive term in the loss function, ensuring consistent importance across different neurons. Second, from an MEI, it is possible to compute an MEI mask that isolates the receptive field (RF) of biological neurons, as described in (Ding et al., 2023).

MEI masks were computed following Ding et al. (2023) by identifying the convex region where pixel values exceed $0.5$ standard deviations. During template learning, applying these masks while calculating the contrastive regularization term of the objective ensures that variations in pixel values are encouraged only within the neuron's RF.

## B    TEMPLATE LEARNING DETAILS

The initialization weights for the fully connected layer in the Implicit Neural Representation (INR) were sampled from a Gaussian distribution with $\sigma = 0.1$. For positional encoding of the pixel coordinates, we used Fourier features with $50$ dimensions with a projection scale of $10$. For the positional encoding of the latent input, we used Fourier features with $50$ dimensions with a projection scale of $0.1$.

During template learning, to train the INR, we employed a jittering grid technique. A grid of $N = 20$ uniformly distributed points $z \in \mathcal{Z}$ along a 1D periodic latent dimension was passed to the INR at each step, generating images $\{g(z)\}_{z \in \mathcal{Z}}$. This grid was jittered at every step of the training procedure to allow the latent input to span the entire range. To define $\mathcal{Z}_+^i$ and $\mathcal{Z}_-^i$ for each point $z_i$, corresponding to near (positive) and distant (negative) points in the latent space, we chose a neighboring radius that spanned $10\%$ of the latent space in each direction, resulting in two near (positive) grid points in each direction for every point $z_i$. All other points were designated as distant (negative). Contrastive regularization was applied solely within the points of the mask obtained from the MEI. The temperature parameter was set to $\tau = 0.3$.

The parameters of the INR were optimized using an Adam optimizer (Diederik, 2014) with a learning rate of $0.001$, focusing on maximizing the activity and contrastive learning objectives as defined in Eq. 1. Training continued for a minimum of $500$ steps and concluded once specific activity requirements were met by the images generated from the jittered grid sampled in the latent space (activity was monitored every $50$ training steps). For experiments involving simulated neurons, the average required activity was set to $\alpha = 0.99\alpha_{MEI}$ and the minimum to $\alpha = 0.98\alpha_{MEI}$. For experiments involving macaque neurons, the average required activity was $\alpha = 0.9\alpha_{MEI}$, and the minimum was $\alpha = 0.85\alpha_{MEI}$, accommodating the expectation that the invariance manifold of biological neurons might not exhibit perfect iso-response properties. The strength of the contrastive regularization term, initially set to $\lambda = 2$, was reduced by a factor of $0.8$ via a scheduler whenever the activity term plateaued (patience parameter of 5). This adaptive schedule, coupled with the condition to satisfy activity requirements to stop training, allows a careful balance between activity maximization and contrastive regularization. As a result, the INR can successfully capture neural invariances across neurons exhibiting varying degrees of invariance.

## C  ROBUSTNESS TO "NEAR" AND "DISTANT" LATENT POINT CONFIGURATION

The contrastive term in the template learning objective (Eq. 1) relies on specifying positive and negative examples for a given latent point $z_i$. Positive examples are images corresponding to latent points $\mathcal{Z}_+^i$ near $z_i$, while negative examples are images corresponding to distant points $\mathcal{Z}_-^i$. This formulation encourages similarity among nearby images and diversity among distant ones, promoting a smooth and diverse learned manifold. Here we assess the effect of "near" and "distant" latent point configuration on the smoothness of the learned manifolds, and their resulting activation. Specifically, we used a complex cell (Fig. S1A) and applied our method with five different settings of "near" vs "distant" range (Fig. S1C): 5%, 10%, 20%, 40%, and 45%. Each value represents the radius of the

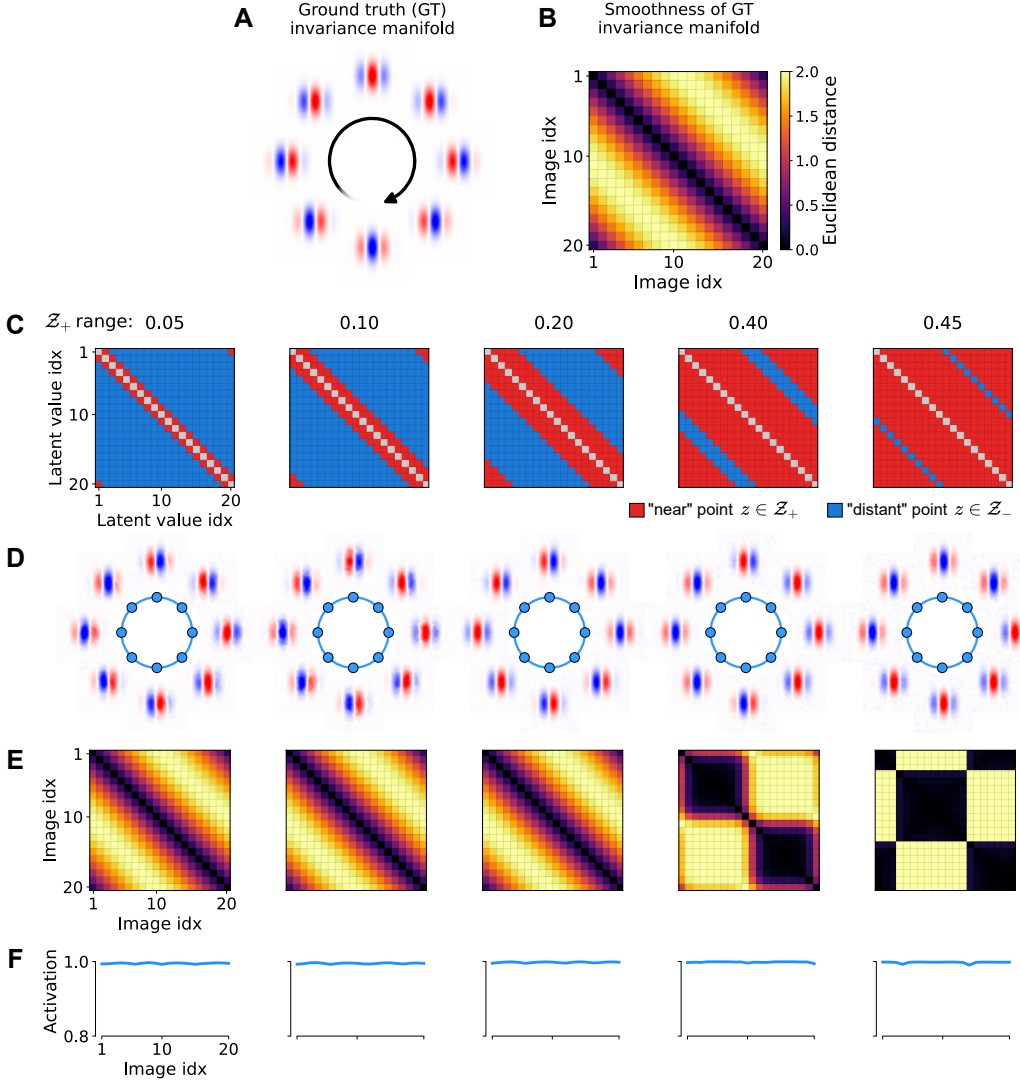

Figure S1: The effect of "near" and "distant" latent point configuration. **A)** Ground truth invariance manifold of a complex cell. **B)** Euclidean distance matrix showing pairwise distances between images sampled from the GT manifold at equidistant phases. **C)** Neighboring mask illustrating "near" (red) and "distant" (blue) points based on varying definitions of proximity ($\mathcal{Z}_+$ range: 0.05–0.45). Each row corresponds to a different $\mathcal{Z}_+$ threshold. **D)** Learned invariance manifolds across different proximity definitions. **E)** Euclidean distance matrices between image pairs sampled from the learned manifolds at equidistant points in the latent space. **F)** Neural activation curves elicited by images sampled from the learned manifolds.

neighboring area as a percentage of the whole latent space. We sampled 20 points from a 1D latent space. Consequently, 5% represents 2 "near" points (1 on the left and 1 on the right) which is the minimum number of "near" points possible. Similarly, 45% represents 18 points (9 on the left and 9 on the right), which is the maximum number of "near" points possible. Our method successfully learns a smooth parametrization of the underlying manifold for most of the "near" percentages considered (Fig. S1D,E). Only in the extreme cases where "near" points represent the majority, smoothness is degraded (Fig. S1D,E last two columns). This is expected, as this solution improves the contrastive score by finding a balance between ensuring high similarity with (roughly half of) the "near" points and maximizing the cosine distance with the (few) "distant" points. Overall, these results suggest that our method presents robustness to the exact setting of "near" vs "distant" point, especially when "near" points represents a small minority.

# D    TEMPLATE MATCHING DETAILS

For each pair of template and target neurons, we performed template matching to maximize the target neuron's response to the template invariance manifold $\tilde{f}_\theta(g_{\phi^*}(A_\psi(\mathbf{x}, \mathbf{y}), z))$ through an affine transformation $A_\psi(\mathbf{x}, \mathbf{y})$.

To minimize the risk of converging on local minima, we initialized the affine transformation by centering the template's invariance manifold on the receptive field of the target neuron, as determined by the centroids of their MEI masks. We then uniformly rescaled the coordinates based on the size ratio between the MEI masks of the target and template neurons. Finally, we rotated the coordinates to the angle that yielded the maximal neural response.

We optimized the parameters of the affine transformation using the Adam optimizer with a learning rate of $0.001$. Training was halted when the average activity of the target neurons ceased to increase, with a patience parameter set to $15$.

To accommodate GPU memory constraints during template matching for a large number of neurons, we batched the procedures into groups of $20$ or $50$ neurons, depending on the available computational resources.

# E    GABOR-BASED SIMULATED NEURON IMPLEMENTATION

We implemented simulated neuron models presenting various invariances using Gabor filters. We obtained simple cells as linear-nonlinear systems using a normalized Gabor filter and a final non-linearity $(\text{ELU} + 1)/2$ which ensures non-negative responses and a unitary maximal response for images with unit norm. To model complex cells, we used a set of normalized Gabor filters with shared parameters but different phases. To achieve phase invariance, the model selects the filter that produces the highest output value, which is then passed through the same final nonlinearity. Similarly, for neurons exhibiting more complex invariances, we obtained sets of filters by combining multiple Gabor filters with parameters that were obtained by varying an underlying 1D continuous variable, and normalizing them. To achieve the desired invariance, analogous to the process used for complex cells, the model selects the highest output value among those generated by the Gabor filters, which is then passed through the same final nonlinearity.

# F    SIMILARITY SCORE BETWEEN MANIFOLDS

We define the similarity score between two manifolds as

$$\text{score}(g_1, g_2) = \frac{1}{2} \left( \frac{1}{n} \sum_{i=1}^{n} \max_{j} \left( \text{sim}(g_1(z_i), g_2(z_j)) \right) + \frac{1}{m} \sum_{j=1}^{m} \max_{i} \left( \text{sim}(g_1(z_i), g_2(z_j)) \right) \right)$$

where $\mathbf{g}_1(z_i)$ and $\mathbf{g}_2(z_j)$ represent the images on the two manifolds, parametrized by the mappings of two INRs, $\mathbf{g}_1$ and $\mathbf{g}_2$. This measure computes the highest cosine similarity between each image on one manifold and the images on the other manifold and vice-versa, then averages these quantities within manifolds and across manifolds to get a measure of similarity between two manifolds.

# G  COMPARISON WITH MODEL-BASED CLUSTERING ALGORITHMS

In this section, we compare our method to the model-based clustering approaches proposed by Klindt et al. (2017) and Ustyuzhaninov et al. (2019). Briefly, both methods rely on DNN-based models composed of two main parts: a core and a readout. The core is implemented as a neural network with 2D convolution layers, extracting nonlinear features across spatial locations. These features are shared across all neurons but are linearly combined by the readout at specific spatial locations to predict each neuron's response. While the core captures shared computations across the neuronal population, the readout's neuron-specific weights, which serve as proxies for neuron-specific computations, can be used to cluster neurons.

Both methods can cluster neurons independently of differences in RF position, as the convolution operation is translation-equivariant. However, the model proposed by Ustyuzhaninov et al. (2019) extends this by incorporating rotation-equivariant convolution layers. This allows the core to learn nonlinear functions not only at different spatial locations but also across different orientations, enabling clustering invariant to both translation and rotation using the feature weights of the readout. While these approaches have proven effective for identifying functional clusters invariant to shifts (Klindt et al., 2017) and rotations (Ustyuzhaninov et al., 2019), our method generalizes these invariances to include all affine transformations, and can straightforwardly be generalized to more powerful transformations as well. Here, we provide an empirical comparison of the clustering results obtained with these different methods.

**Simulated neurons**  To keep this experiment focused on assessing how well our approach accounts for transformations beyond translation (Klindt et al., 2017) and rotation (Ustyuzhaninov et al., 2019), we considered two types of neurons: odd and even simple Gabor cells. For each type, we considered two different RF sizes, two different orientations, and five randomly sampled locations, giving rise to 20 simulated neurons per cell type, and 40 in total (Fig. S2A). To keep the demonstration and its results clear, we focus on our method's ability to cluster neurons in a way that is invariant against transformations of pixel coordinates and did not demonstrate another ability of our model: to cluster by invariances.

**Implementation of model-based methods**  We implemented the methods proposed by Klindt et al. (2017) and Ustyuzhaninov et al. (2019) using convolutional and rotation-equivariant convolutional models (using steerable filters as in Ecker et al. (2018)), respectively. Given the linear-nonlinear nature of the cells considered (implementation details in AppendixE), we used a single-layer core followed by a Gaussian readout as in Lurz et al. (2020) and a $(\text{ELU} + 1)/2$ final nonlinearity. The Gaussian readout is a special case of the factorized readout which was used by Klindt et al. (2017). Specifically, while the factorized readout learns a spatial mask for each neuron to learn relevant spatial locations (i.e., neuron's RF) for predicting neural responses, the Gaussian readout learns a single location in the spatial dimensions for each neuron and linearly combines features across channels at that specific location. Thus the Gaussian readout is a factorized readout where the spatial mask is reduced to a single pixel on the output tensor of the core. This results in a significant reduction in the number of readout parameters and has been shown to be more data-efficient (Lurz et al., 2020).

To minimize redundancies in features learned by the core, we enforced readout feature weights to be positive and adjusted the model capacity by fixing the number of channels to eight for the convolutional model (corresponding to cell type $\times$ number of sizes $\times$ number of orientations) and four for the rotation-equivariant convolutional model (corresponding to cell type $\times$ number of sizes). For the rotation-equivariant model, we uniformly discretized the full $2\pi$ range of orientations into four possible values $\in \{0, \pi/2, \pi, 3\pi/2\}$, matching the distance between the orientations considered in our simulated population (i.e., 0 and $\pi/2$). To further avoid redundancies and simplify training, we fixed the readout position for each neuron to its correct location, to focus the training of these models on learning the convolutional kernels and neuron-specific feature weights.

For the remainder of this section, and in Fig. S2, we refer to the convolutional (Klindt et al., 2017) and rotation-equivariant convolutional (Ustyuzhaninov et al., 2019) models as CNN-based and RotEqCNN-based, respectively.

**Training of model-based methods**  Both models were trained on 150,000 white noise images presented once, grouped in batches of size 10, minimizing the mean squared error between the ground truth responses of our simulated simple cell models and the predictions of the convolutional architectures. Similar to Klindt et al. (2017), to encourage the models to learn distinct kernels and

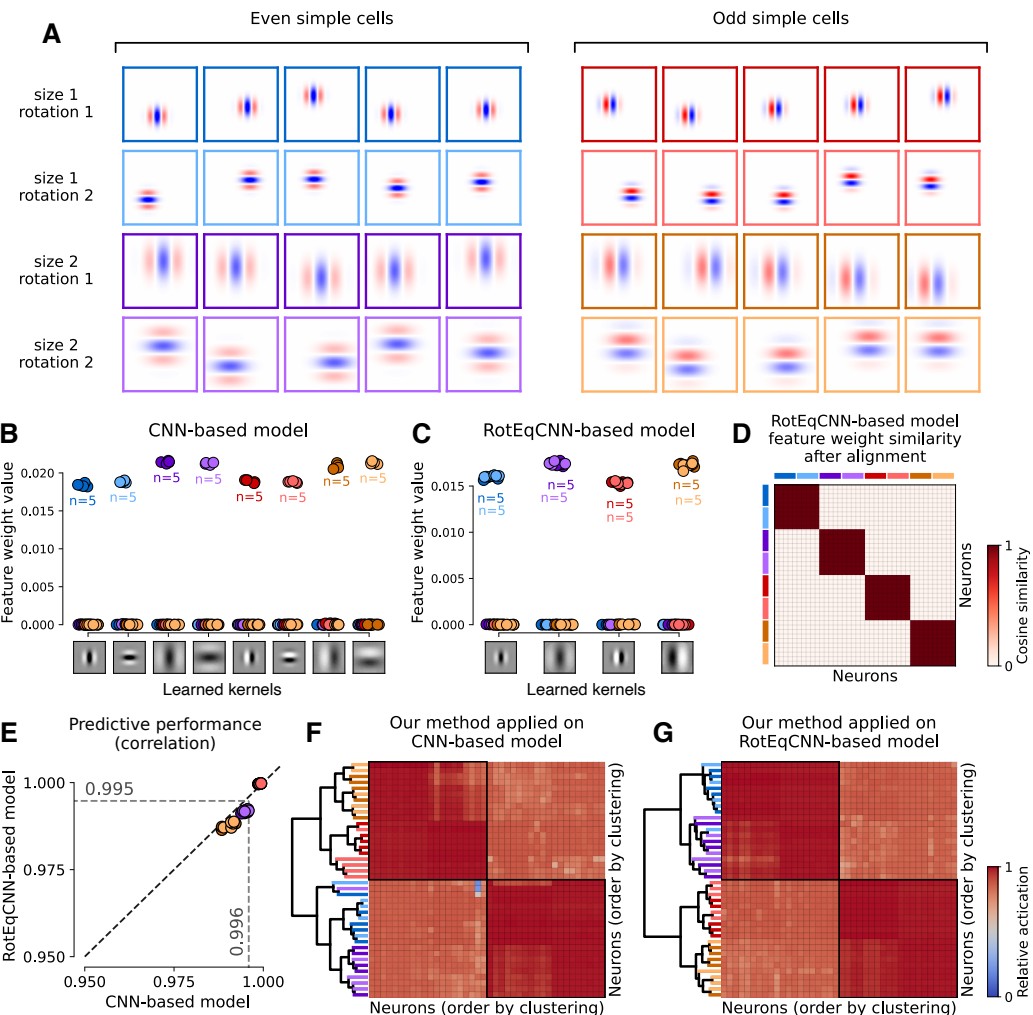

Figure S2: Comparison with model-based clustering algorithms. **A)** Simulated neuron population, containing odd and even simple cells with 2 different sizes, 2 different orientations, and 5 different randomly sampled positions. **B)** Feature weights of CNN-based model (Klindt et al., 2017).The x-axis represents the learned kernels, while the y-axis shows the corresponding feature weights for each neuron. Colors correspond to neurons colors in panel A. Neurons with the same neuron type, rotation, and size can be clustered together as they present similar feature weights. **C)** Feature weights of RotEqCNN-based model Ustyuzhaninov et al. (2019), while disregarding the specific orientation, by marginalizing (i.e., summing) the weights across different rotations of the same kernel. The x-axis represents the learned kernels, while the y-axis shows the corresponding marginalized feature weights for each neuron. Colors correspond to neurons colors in panel A. **D)** Cosine similarity between feature weights after aligning the weights regardless of their orientation. Neurons belonging to the same type and sharing size have high cosine similarity. Colors on the top and on the left are the same as in panel A. **E)** Scatter plot showing predictive performance (in terms of correlation to target responses) of CNN-based model vs RotEqCNN-based model. **F)** Resulting clustered activation matrix from the application of our method on the CNN-based model. Dendogram colors are the same as in panel A. **G)** Resulting clustered activation matrix from the application of our method on the RotEqCNN-based model. Dendogram colors are the same as in panel A.

allow clustering of neurons based on distinct functional properties, we applied an L1 regularization to the readout feature weights with a strength coefficient of $0.01$. This setup enabled the models to effectively capture the functional diversity of the neuron population while maintaining both interpretability and efficiency. For training, we used the Adam optimizer with a learning rate of $0.01$.

**Clustering via model-based methods** Both models achieved high predictive performance, computed as the correlation between the ground truth responses of our simulated simple cell models and the predictions of the trained models to newly sampled white noise images (see Fig. S2E), and learned to associate different neurons to different channels, grouping them according to the design of their architecture. Specifically Fig. S2B shows how, at the end of our training of the CNN-based model, neurons of the same type, size, and orientation learn similar feature weight vectors independently of their position. Consequently, while clustering based on these feature weights is invariant to differences in location, it does not disregard differences in size and orientation.

Similarly, Fig. S2C and D show how, at the end of our training of the RotEqCNN-based model, neurons can be clustered using their feature weights according to differences in type (odd vs even) and size, disregarding differences in orientation and position. Specifically, neurons performing the same computation but with different orientations picked the same kernel (i.e., the same computation), which in the RotEqCNN-based model is present with different rotations. In this case, a simple marginalization based on summing feature weights across the different rotations is sufficient to cluster neurons with unique type and RF size, disregarding RF location and orientation (Fig. S2C).

Additionally, we performed an alignment procedure similar to Ustyuzhaninov et al. (2019) to compare feature weights irrespective of rotational differences. By finding the optimal rotation that aligns the feature weights of two neurons, we can assess their computational similarity while disregarding orientation. The results, shown in Fig. S2D, highlight that neurons of the same type and size exhibit high cosine similarity in their aligned feature vectors, where the optimal rotation was found by minimizing the Euclidean distance between weights. This confirms the ability of the method introduced by Ustyuzhaninov et al. (2019) in clustering neurons despite variations in receptive field (RF) location and orientation. However, similar to Klindt et al. (2017), this method remains limited in its ability to cluster neurons irrespective of other nuisance RF properties such as size, which is known to vary with eccentricity in higher mammals' visual systems (Wilson & Sherman, 1976; Cavanaugh et al., 2002; Harvey & Dumoulin, 2011).

**Comparison with our method** Lastly, we applied our method, activation-based clustering via template learning and template matching, to both models. The clustering results are consistent across both architectures (Fig. S2F,G) and group neurons with the same type (odd vs even) together, successfully disregarding differences in RF position, orientation, and size (further confirming the results shown in Fig. 2). Importantly, the consistency of our results across both architectures highlights the model-agnostic nature of our method. That is, as long as an accurate response-predicting model is used our method yields consistent clusters, which is a crucial advantage in not constraining (or biasing) such findings or analyses to specific architectures or their implementation. An additional important observation is that the activation matrix, which contains the activations after template matching, presents relatively high values between neurons of different types. This reflects the fact that the neuron types considered, odd and even simple cells, differing only in phase, perform quite similar computations, as when properly shifted odd Gabor patterns can activate an even simple cell relatively high, and vice versa. In contrast, this computational similarity is not visible in the feature weights of the model-based methods.

## H MACAQUE RESPONSE-PREDICTING MODELS DETAILS

In our experiments on macaque V1 neurons, we employed the model used by Baroni et al. (2023) to predict neural responses to 93×93 pixel grayscale images, representing 2.65 degrees of the visual field. This model comprises an ensemble of three DNN-based models, each trained separately to minimize the Poisson loss between predicted and recorded neural responses. The final predictions from the ensemble model are obtained by averaging the outputs of individual models.

Each DNN-based model is composed of two main parts: a core and a readout. The core is implemented as a neural network with 2D convolution layers, extracting nonlinear features across spatial locations. These features are shared across all neurons but are linearly combined by the readout at neuron-specific spatial locations, with the resulting values passed through a final ELU + 1 nonlinearity, to predict each neuron's response. More specifically, the core consists of three layers of depthwise-separable convolutions, each with a kernel size of 24, 9, and 9, respectively, 32 feature channels per layer, followed by batch normalization, ELU nonlinearity, and, for every layer except the first one, a squeeze-and-excitation block. To predict single-neuron responses, each DNN-based model employs a pyramidal readout (Sinz et al., 2018) followed by an ELU + 1 nonlinearity.

To assess the generalization of our method across different models, we trained a second ensemble model. Similar to the first ensemble model, this ensemble model includes three separately trained DNN-based models. However, in contrast, for each DNN-based model we used the Gaussian readout introduced byLurz et al. (2020). We also did not use the squeeze-and-excitation blocks, while the other aspects of the models remained the same.

Following the training approach of Baroni et al. (2023), we z-scored the stimuli before training each model in the ensemble using the Adam optimizer (Diederik, 2014) with an initial learning rate of 0.004, a batch size of 128, and a learning rate reduction by a factor of 0.3 up to three times when there was no improvement in validation loss for three consecutive epochs. This second ensemble architecture achieved strong predictive performance, with a correlation of 0.71 between predicted and observed trial-averaged responses, comparable to the predictive performance of the first ensemble, which achieved a correlation of 0.73.

# I  SUPPLEMENTARY FIGURES

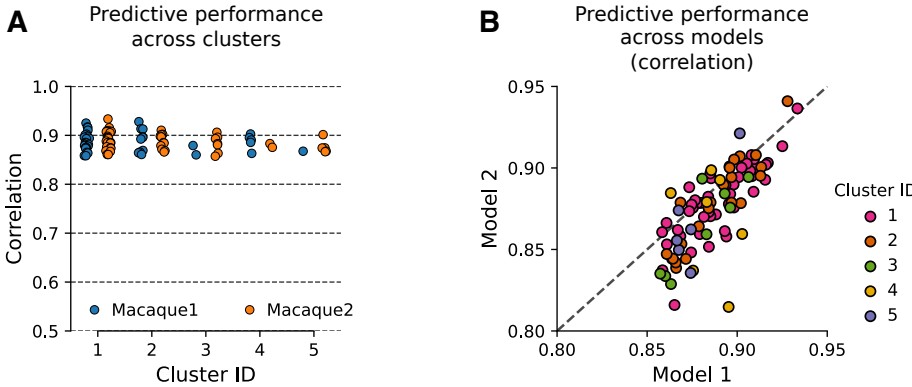

Figure S3: Predictive performance of response-predicting models. **A)** Predictive performance of Model 1 (which was used for the main results) in terms of correlation to trial-averaged responses on left out images (*i.e.*, test set). **B)** Predictive performance of Model 1 vs Model 2 in terms of correlation to trial-averaged responses on left out images (*i.e.*, test set). Model 2 has a different architecture but was trained on the same set of neurons as Model 1.

