# OpenReview forum: "Learning and aligning single-neuron invariance manifolds in visual cortex"
_ICLR.cc/2025/Conference — ICLR 2025 Oral_

### Official Review · Reviewer_aC4g · 2024-10-31

**Soundness:** 4
**Presentation:** 4
**Contribution:** 3
**Rating:** 8
**Confidence:** 5

**Summary:**

This paper proposes a new method to map invariance manifolds of visual neurons. They combine prior methods to arrive at a new formulation that allows for continuous deformations of MEIs and clustering that marginalizes over affine spatial transformation. They validate their method on simulations and test it on Macaque V1 data.

**Strengths:**

The paper is well written, cites relevant prior literature and does a good job working through the logic and experiments. Figure 1 is very illustrative and of high quality. The result figures are also very clean and show convincing results. I like the focus on continuous transformations, allowed by implicit representations, rather than pointwise estimates such as Cadena et al. 2018.

The 'new cell types' on V1 data are exciting, looking forward to closed loop validation.

**Weaknesses:**

It would be instructive to discuss the concept of a nullspace, as being the input space that the neural response is invariant to. One can think of an MEI as a first order approximation to the response function. Thus, for such a linear function it is quite natural to think about its nullspace aka the invariance manifold. This actually leads to a question: there seems to be a way to game the objective function. One could simply keep the MEI in the RF identical while making large changes outside the RF (for instance, strongly changing backgrounds); How is this avoided?

108: The method in Klindt et al. 2017 performance functional clustering that is invariant to RF position; The method in Ustyuzhaninov et al. 2019 extends this to clustering that is also invariant to orientation; Arguably, the main difference that the affine transformations in this paper add on top of those prior methods are: scaling, shearing and reflections of MEIs

Invariance is a broader concept than just invariance around the MEI. One can have iso-response input manifolds showing different invariances at every activation level of the neuron. Please add this distinction and explain why you only study invariance at the MEI.

It would be great to add comparisons to prior methods. For instance, your method should be able to solve the functional clustering despite RF shifts like in Klindt et al. 2017; It should also be able to solve the functional clustering despite RF shifts and rotations like in Ustyuzhaninov et al. 2019. I am happy to raise my score if you can add those comparisons.

**Questions:**

117-119: please clarify

Why do you use a periodic 1D latent variable? Please justify and explain

152: why remove batchnorm?

153: steps = typo?

155: Are the Fourier features new compared to Baroni? They are pretty standard in Sirens / implicit function modelling. Why do they allow better controlling spatial frequency etc., do you have data to support this?

Why did you choose cosine similarity as image metric? Might this have shortcomings?

253: neuron + s

Why don't you use the activation matrix directly as a similarity matrix input to agglomerative clustering, why first compute the cos.sim. between rows? [more data massaging -> more skeptical]

---

> ### Author Response · Authors · 2024-11-14
> **One clarifying question to Reviewer aC4g**
>
> Thank you for your insightful review and feedback. We believe we can address most of your concerns, but we have one quick clarifying question to ensure we fully understand your concern. In particular, regarding comparisons with Klindt et al. (2017) and Ustyuzhaninov et al. (2019):
>
> If we understood correctly, you’re suggesting adding a comparison showing explicitly that our method is able to identify clusters despite differences in RF position (as in Klindt et al. 2017) and orientation (as in Ustyuzhaninov et al. 2019). We believe our results already show this: we simulated neurons with similar invariances but spatially transformed RFs using different affine transformations, including differences not only in shift and rotation but additionally size and shear (Fig 2A bottom row). Qualitatively (Fig 2B) and quantitatively (Fig 2D,E), our method captures both the RF pattern and its transformations along the manifold, as well as the underlying affine transformations, showing robustness in accounting for spatial variations beyond those in Klindt et al. (2017) and Ustyuzhaninov et al. (2019).
>
> However, it could be that we did not understand the exact issue you are raising. We would therefore be grateful if you could clarify what you felt is missing from these analyses. Thank you again for your input!

---

> > ### Comment · Reviewer_aC4g · 2024-11-14
> > **Clarification re additional experiments**
> >
> > My apologies, the comment was a bit unclear. I am suggesting running those other two methods as comparisons, e.g., on three settings of increasing complexity:
> >
> > 1) Only spatial translations: Klindt17 works, Ustyuzhaninov19 works, current submission works
> > 2) Spatial translations and rotations: Klindt17 fails, Ustyuzhaninov19 works, current submission works
> > 3) Full affine transformations: Klindt17 fails, Ustyuzhaninov19 fails, current submission works
> >
> > That would be a nice model comparison and demonstration how the proposed method goes beyond prior art.

---

> > > ### Author Response · Authors · 2024-11-15
> > > **Follow up on the clarification re additional experiments**
> > >
> > > Thank you very much for the quick response and clarification. We now understand that you’re proposing a structured comparison across different levels of affine transformation complexity to explicitly demonstrate where our method outperforms Klindt et al. (2017) and Ustyuzhaninov et al. (2019).
> > >
> > > On one hand, Figure 2 of our paper already demonstrates that our method successfully accounts for the highest level of complexity you described. On the other hand, the fact that previous methods do not handle full affine transformation differences among cells of the same type is a limitation implicit in their design. For this reason, we feel it may add limited value to empirically demonstrate what we theoretically understand about the limitations of these approaches (and which you also accurately anticipated in your response). That said, we recognize the appeal of performing such an analysis, particularly as it would demonstrate that our approach is model-agnostic: as long as one uses an accurate response-predicting model, our method yields consistent clusters.
> > >
> > > While performing this analysis might be challenging within the limited rebuttal period—including implementing the models, training them, and addressing potential redundancies in their weight vectors used for clustering—we will make every effort to conduct this, or a similar, analysis. We hope this addresses your request while also highlighting the broader advantages of our approach.

---

> ### Author Response · Authors · 2024-11-21
> **Authors response to reviewer aC4g (part 1)**
>
> We would like to thank you for your detailed and thoughtful review. We are glad that you appreciated the clarity of our work, the quality of its figures, and our focus on continuous transformations. We also appreciate your insightful suggestions and address each of your points below.
>
> **Re: Discussion of nullspace and is there a way to game the objective function?**
>
> Neurons are generally insensitive to large portions of the visual field (in our case, the input image) outside of their receptive field (RF). These areas could be considered as belonging to the “nullspace” of the neuron. If we understand your question correctly, you’re suggesting that maximally exciting images (MEIs) could potentially be modified in these regions without affecting the neural response, thereby potentially gaming the objective function.
> Our method incorporates two mechanisms to specifically prevent this. First, the combination of a norm constraint (see section 3.1 Template Learning under Objective function) with the activity maximization objective ensures that features (i.e. pixel variations) are concentrated only in regions relevant to the neuron. Because neurons “like” contrast, this constraint makes the method put contrast only where it gets activation from the neuron. It is important that the selected norm value is low enough to prevent pixel saturation within the RF and to avoid the appearance of random artifacts outside the RF. This mechanism alone should be sufficient to avoid features to appear outside of the RF of the neuron.
> Second, in addition, we apply the contrastive regularization terms only within the RF of the neuron using a mask computed from the MEI which as well satisfies the norm constraint (see appendix B, Template Learning Details). This ensures that the contrastive regularization term does not encourage features to appear in the neuron’s nullspace. Taken together these mechanisms ensure that the focus remains on regions of the input space that actually drive neurons' response.
>
> **Re: Comparison with Klindt at al. (2017) and Ustyuzhaninov et al. (2019)**
>
> Klindt at al. 2017 and Ustyuzhaninov et al. 2019 propose to cluster neurons based on the feature weights of the readout of each neuron. Klindt at al. 2017 proposes a method that divides neurons in functional clusters that are obtained up to differences in RF position, Ustyuzhaninov et al. 2019 extend this to rotations. Our method finds clusters that extend this property to the remaining affine transformations. Few additional differences are present between our method and theirs:
> First, our method is model agnostic, whereas their method is model dependent, meaning that while our method can be applied to any accurate response predicting model of neurons their method necessitates to obtain an accurate response predicting model using a specific type of model architecture (e.g. Ustyuzhaninov needs a rotation equivariant core and could not straightforwardly be applied to a transformer).
> Second, our method aims at clustering neurons specifically comparing their invariances, whereas their method aims at clustering neurons according to the whole response function. Whether this is to be preferred or not, might depend on the questions that are being asked. It is worth pointing out that in a situation where two neuron clusters differ in invariance, our method will clearly show that the differentiating factor is the invariance and how it is different, whereas the Ustyuzhaninov et al. 2019 method will just indicate that the two clusters of neurons are different but not directly offer insights on how.
> Third, in our case the metric used for template alignment and neural clustering is neural activity, which has an intuitive interpretation. In contrast, the readout weight vectors used by their method to perform clustering are not immediately interpretable in terms of functional properties, mainly because distances in the space of readout feature weights might not be representative of distances in terms of neural functional properties. This is strongly related to the issue of feature redundancies mentioned in the related work section for which you ask for clarification (see reply to the next point).

---

> ### Author Response · Authors · 2024-11-21
> **Authors response to reviewer aC4g (part 2)**
>
> **Re: 117-119 please clarify**:
>
> In Ustyuzhaninov et al. 2019, clustering relies on a model with a core+readout architecture. This architecture incorporates equivariance to shifts and rotations, with the readout combining features from different channels at specific spatial location and orientation using a vector of feature weights for each neuron. Clustering is performed based on these feature weight vectors, which can be considered a fingerprint of the neuron’s properties.
> The issue of redundancy arises as the same neuronal response function can be realized by different combinations of  channels of the core. Consequently, two neurons with very similar properties might present very different fingerprints (weight vectors) and could therefore be grouped into different clusters. The severity of this issue is closely tied to the channel dimensionality of the core’s output (i.e. the model capacity), and while it might be mitigated by some regularization approach or pruning, it still remains a non-trivial issue to address. Since our clustering method is model-agnostic and based on neuron responses rather than readout weights, it is not subject to this issue. In the updated version of the manuscript, we revised the text to make the explanation of this point clearer.
>
> **Re: Empirical comparison of our method to Klindt at al. (2017) and Ustyuzhaninov et al. (2019):** Regarding the analysis you requested, we are working on it and will update you once the results are ready.
>
>  **Re: Invariance is a broader concept than just invariance around the MEI**
>
> Thank you for this remark. A similar point was raised by another reviewer (5DFt), and we agree that it would be beneficial to expand on the justification behind our choice of focusing on invariances over maximally exciting stimuli only. It would indeed be interesting to explore a neuron’s invariances at different response levels, as this could provide a more comprehensive characterization of the entire response function of the selected neuron. However, capturing invariances at lower-than-optimal response levels presents challenges that go beyond the scope of this study. For instance, images that both satisfy the norm constraint and achieve the target activation level could be generated in two ways: 1) by learning patterns only within the RF, which is the desired outcome, or 2) keeping the pattern within the RF similar to the case of peak activation but with a lower norm and “cheat” the norm constraint by introducing additional patterns outside the neuron’s RF. In the latter case, the features outside the RF represent uninformative contrast allocations unrelated to the neuron’s function and are merely artifacts of the procedure. Resolving this issue would require the development of additional mechanisms. Furthermore, the invariance manifolds are likely to become very high-dimensional and exhibit a complex topology, significantly increasing the difficulty of capturing them effectively. Due to these challenges, exploring invariances at  suboptimal activation levels extends beyond the scope of this work. Our focus on invariances at (or close to) the peak activation is motivated by three factors. First, this approach aligns with how invariances are often described in neuroscientific research. For instance, when one says that a complex cell present phase invariance, it is assumed that it is selective to Gabor stimuli with specific parameters and it is invariant to changes in phase of Gabor stimuli with specifically these parameters (i.e. the stimuli for it is particularly selective to). Second, investigating stimuli (and invariances) at peak activation is fundamental to understanding neural coding, as it reveals the features to which neurons are most selective and, therefore, offers critical insights into their functional role. Third, invariance manifolds at peak activation are likely to be lower dimensional, which makes their identification a more constrained yet meaningful problem, that enables us to focus on the alignment aspect of our approach for population-wide comparisons of single-neuron invariance manifolds. In the updated version of the manuscript, we ensured to specify that our method focuses specifically on the invariances at peak activation level, justifying the reasons behind our choice.
>
> **Re: Why do you use periodic 1D?**
>
> Given the focus of our work on V1, we chose a 1D latent space as we expected to mainly identify neurons with no invariances (simple cells) or a one-dimensional phase invariance (complex cells). Since phase invariance is periodic, we designed the latent space of our INR to be periodic as well.
> Exploring the dimensionality of neurons belonging to the non-gabor clusters with higher dimensional latent space (as suggested in Baroni et al. 2023) is a natural extension but was not the focus of our  current work.

---

> > ### Author Response · Authors · 2024-11-21
> > **Authors response to reviewer aC4g (part 3)**
> >
> > **Re: Why remove batchnorm from Baroni et al. 2023?**
> >
> > Initially, we opted for not using batch normalization as we believed it might interfere with the template matching step when pixel coordinate statistics change due to the affine transformation. However, we later realized this is not an issue since, during template matching, the parameters of the INR, including the batch normalization parameters, are fixed. Nonetheless, as the method performs equally well without batch normalization, we will retain the current architecture for its simplicity. We revised the main text accordingly.
> >
> > **Re: Are the Fourier features new compared to Baroni et al. 2023 and do you have data to support the controlling spatial frequency?**
> >
> > Baroni et al. 2023 already utilized Fourier features on pixel coordinates. Since the latent input $z$ is also an input to the INR, we extended the use of Fourier features to $z$ as well. While we have not conducted a systematic analysis of their impact on controlling the spatial frequency of the output, this effect has been studied and well-documented in the literature (see, for instance, [this explanation](https://bmild.github.io/fourfeat/) and the corresponding paper: [Tancik et al., 2020](https://arxiv.org/abs/2006.10739)).
> >
> > **Re: Why don't you use the activation matrix directly as a similarity matrix input to agglomerative clustering and why cosine similarity?**
> >
> > We believe there might have been a misunderstanding here. We did use the entire activation matrix directly as the input matrix to the agglomerative clustering method. However the metric we employed for the clustering process is cosine dissimilarity (1 - cosine similarity) which effectively computes the cosine similarity of the matrix rows. Alternatively, one could use Euclidean distance to measure the difference between the rows of the activation matrix. However, for rows belonging to neurons belonging to types represented to varying degrees in the population, Euclidean distance misrepresents similarity between such neurons. This happens because Euclidean distance depends on the magnitude and number of non-zero elements in the vectors. Cosine similarity, however, considers only the angular relationship and treats all orthogonal vectors equally, regardless of their magnitude (i.e. number of non-zero elements). This makes it more appropriate for clustering sparse activation vectors. Let’s consider for instance the example of two neurons A and B that are very different not only from each other but also from a group of neurons C that share very similar invariance properties.  Intuitively, neurons A and B should not be grouped with any other neurons and should form isolated “clusters”. In this example, after alignment, neurons A and B are represented by two almost orthogonal very sparse activation vectors (i.e. their invariance manifold highly excites the neuron itself but results in low activation for other neurons), whereas neurons C are represented by a vector of large magnitude orthogonal to A and B. In this example, while cosine dissimilarity correctly identifies that all neuron types are orthogonal and equally distant, Euclidean distance suggests that the distance between neuron A and B is smaller than their distance to neurons C, therefore incorrectly biasing neurons A and B to be grouped together. We have made the explicit mention of cosine dissimilarity as a distance metric for clustering more prominent in the text.

---

> ### Author Response · Authors · 2024-11-24
> **Follow up on the additional experiments**
>
> We are happy to let you know that we managed to perform the analysis you requested, comparing our method to that of Klindt et al. (2017) and Ustyuzhaninov et al. (2019), and have included it in the new revision of our manuscript (Appendix H). The results empirically demonstrate the flexibility of our approach in accounting for affine transformations (similar to the results in Figure 2 of the initial submission), while highlighting that the other two methods can only account for shifts (Klindt et al. 2017) and rotations (Ustyuzhaninov et al. 2019). Importantly, we applied our approach on both architectures from Klindt et al. (2017) and Ustyuzhaninov et al. (2019), and show that on both models our method yields consistent clusters, highlighting the model-agnostic nature of our approach.
>
> Thank you again for your valuable input, and we look forward to hearing your feedback on both this new analysis and our earlier clarifications.

---

> > ### Comment · Reviewer_aC4g · 2024-11-25
> > **Acknowledgement**
> >
> > Thanks for the additional experiments. I have raised my score. Please go through the new Appendix section again and check for grammar and typos, also please add a description to the caption of Fig. S2B to make it clear that distinct colors represent distinct readout weights. Thanks!

---

> > > ### Author Response · Authors · 2024-11-25
> > > **Thank you!**
> > >
> > > Thank you for your response and for raising your score! We will carefully review the new Appendix section for grammar and typos and ensure the captions of Fig. S2B,C clearly describe the meaning of the distinct colors.
> > > Thanks again for your thoughtful comments!

---

### Official Review · Reviewer_Nobq · 2024-11-01

**Soundness:** 3
**Presentation:** 3
**Contribution:** 3
**Rating:** 8
**Confidence:** 2

**Summary:**

This paper proposes a method for categorizing the receptive fields of visual neurons into classes that are invariant to affine transformations. For a given neuron (referred to as the template neuron), they first employ a previously developed method (Baroni et al. 2023) to identify a one-dimensional manifold of images that maximize the response of the neuron (referred to as the template manifold). Then, given another neuron (referred to as target neuron), they identify an affine transformation of the template manifold that maximizes the response of the target neuron. If the transformed template manifold produces strong responses in the target neuron (and the converse holds), then the template neuron and target neuron are identified as belonging to the same "class". They apply their method to synthetic neurons and biological neurons to identify classes of neurons with receptive fields that are "equivalent" up to affine transformation.

**Strengths:**

Overall I enjoyed reading this paper. It provides a systematic method for classifying neurons by their receptive fields. As the authors show, the method captures previously identified receptive fields in V1 (simple cells and complex cells) can be used to identify and characterize previously unidentified receptive fields in V1. The general procedure seems promising.

**Weaknesses:**

As someone who is not familiar with these types of work, it took me some time to appreciate what this paper is doing. For example, the authors state early on that they are seeking a method that "should allow for a meaningful measure of similarity between two invariance manifolds". I had no idea what this meant when I first read it. There was no mention of what this "invariance manifold" is prior to that statement (it's mentioned in the abstract, but not the introduction). I think that slowly walking through an example (maybe with a figure) with simple cells would be helpful.

A few other comments: While invariance to affine transformation seems reasonable in V1, it's not clear how one would want to define invariance in high-order visual areas, though I don't see this as an issue that needs to be addressed in this work.

This paper relies on existing methods which are only briefly explained. (I'm not familiar with these works, hence my low confidence score.)

**Questions:**

- How are $\mathcal{Z}_+$ and $\mathcal{Z}_-$ defined? It seems like this could have a significant effect on the learned manifold of stimuli.

---

> ### Author Response · Authors · 2024-11-21
> **Authors response to reviewer Nobq**
>
> We appreciate your positive feedback, recognizing the potential of our approach in classifying neural invariances and uncovering new insights into V1 neurons, and we are thankful for your constructive suggestions to improve clarity and accessibility of our submission.
>
>
> **Re: Clearer introduction of the term “invariance manifold”:**
>
>  Thank you for pointing out the need for a clearer introduction of this concept. In the revised manuscript, we ensured to define the term “invariance manifold” explicitly before using it and included a concrete example to provide additional context and clarity for readers unfamiliar with the concept.
>
> **Re: Affine transformations are reasonable in V1, what would be a suitable choice in high-order visual areas?**
>
> This is an intriguing question. In V1, it is well established that neurons perform similar computations across varying locations, orientations, and scales—differences that can be described by affine transformations. This understanding directly motivates our approach, which seeks to ‘filter out’ these established functional variations, allowing for a comparison of neuron invariance properties beyond these aspects. While functional organizational principles such as retinotopic mapping and eccentricity-related scaling of receptive field (RF) size are thought to persist after V1, it remains an open question which specific stimuli these neurons are selective to and what invariance transformations connect these stimuli.  One option would be to proceed iteratively, first investigating the code with methods such as ours, identifying the most significant transformations along which the code varies between neurons, incorporating these back into the methods in order to ‘filter them out’ and hence discovering further transformations, and so on. But as you already indicate, these broader questions are beyond the scope of the current work.
>
> **Re: This paper relies on existing methods which are only briefly explained**
>
>  To address this concern, we expanded the related work section and improved the method section ensuring that it is now self-contained and accessible and that it does not rely on familiarity of the reader on past work.
>
> **Re: Definition of $\mathcal{Z}+ and \mathcal{Z}-$ and discussion of their importance**
>
> Thank you for raising this question. It seems that how near and distant points are defined in the latent space requires further clarification, as this was also noted by another reviewer. In the appendix, we stated that we chose a neighboring radius spanning 10% of the latent space in each direction. This means that if a grid of 20 points is sampled from the latent space during training, 2 points in each direction are considered “near,” while the rest are classified as “distant.” However, the method is robust to the exact criteria used for classifying near and distant points, as long as the classification is reasonable.  In the revised version of the manuscript, we improved the corresponding paragraph (Appendix B) to provide greater clarity of the definitions of ‘near’ and ‘distant’ points. Additionally, we included an analysis (Appendix G) to demonstrate the method’s robustness to the specific details of their definition.

---

> > ### Comment · Reviewer_Nobq · 2024-11-25
> >
> > Thank you for your thoughtful response and for your clarifications. I maintain my positive score.

---

> > > ### Author Response · Authors · 2024-11-25
> > > **Thank you!**
> > >
> > > Thank you for your response and for taking the time to review our clarifications. We appreciate your positive evaluation and thoughtful feedback.

---

### Official Review · Reviewer_5DFt · 2024-11-04

**Soundness:** 3
**Presentation:** 4
**Contribution:** 3
**Rating:** 8
**Confidence:** 5

**Summary:**

The authors developed a method to learn the invariant manifold of a visual neuron / encoding model of a visual neuron. Specifically, they used implicit neural representation to parametrize images and uses a circular variable to condition the image parametrization, creating a 1d circular submanifold. Then they optimized the image parametrization via activation maximization and contrastive loss to find the invariance manifold for a given neuron. Additional template matching step through fitting affine transforms allowed the method to account for difference in receptive field size location / rotation allowing for comparison on the population level. They validated the method on synthetic neuronal data and applied it to real neuronal data from monkey V1, finding clustering across neurons.

**Strengths:**

- The problem of characterizing neuronal invariances systematically in a modern way is important and under-explored.
- The method is well presented and principled, and innovative in combining INR, contrastive learning, affine fitting in an elegant way. Similar ideas can potentially impact many other fields.
- Overall, the presentation of the results are stunning and visually pleasing. Kudos on your efforts for making these.
- The method is very thoroughly evaluated on synthetic and real neural data, including consistency across noise seed, robustness across encoding models etc.

**Weaknesses:**

- Intuitively and empirically, the closer to the MEI / peak, the “smaller” the invariant manifold [1]. One weakness is the proposed objective is that it didn’t explicitly allow setting a “target level” of $\alpha_i/\alpha_{MEI}$ but seek to optimize the activation value (→ 1). This could result in the invariant manifold being too close to the MEI and not having enough variability, which seems to me be the case from Fig. 2.
A potential simple improvement will be to allow a target level to be set like having $|\alpha_i/\alpha_{MEI}-\lambda|$ in the objective. c.f. Eq.2 in [1].

[1]: Wang, Ponce, On the Level Sets and Invariance of Neural Tuning Landscapes, 2022, PMLR, https://arxiv.org/pdf/2212.13285

**Questions:**

- One related line of work in this domain the authors seem to miss to compare / discuss is [1, 2]. There the authors used deep generative models $G:z→I$ to parametrize natural images. After finding the activation maximizing latent $z^*$, they sampled images from a 2d submanifold around $z^*$ from the latent space of $z$, and directly find the 1d invariance / iso-response manifold (S1) from this 2d slice. They also measured this invariance manifold on various target levels, where the authors seem to miss.
So it seems like a comparable method to find invariance transformations for image computable neuron model, that is also highly activating.
Though targeting slightly different questions, their work seems related and worth mentioning.

[2]: Wang, Ponce, Tuning landscapes of the ventral stream, 2022, Cell Rep.

- Intuitively, the contrastive loss seems a little redundant, do we really need the numerator positive pair part? e.g. it feels like purely by the smoothness of image parametrization, the near by latent should give rise to the similar images. We just need to make distant latent to give distant images.
( Is it there for numerical stability purpose? )
- Necessity of “Invariance alignment matrix”
For Fig 2CD and 3B, alignment of invariant manifolds, how much of it comes from the similarity of the MEI themselves (applying the fitted affine transformed), how much comes from the invariance manifold?
It seems the images from invariant manifold are all “variation on a theme”, i.e. all pretty close to the MEI, so I think MEI will give rise to similar plots as Fig2C,Fig3B. Can you compare the matrix for MEI and the matrix for invariance manifold?
Given so what’s the additional information in the “alignment of invariance manifold”?
- The method is amazing, but what else can the proposed method contribute to visual neuroscience beyond defining cell type clusters?
- In the end, the proposed method can only find 1 dim invariance manifold. Though we know the neurons can be invariant to much higher dim inv manifold (could be as high as d-1 dim, where d is stimuli dim).
Can this method be used to figure out the true dimensionality of that manifold?

---

> ### Author Response · Authors · 2024-11-21
> **Authors response to reviewer 5DFt (part 1)**
>
> Thank you for your insightful feedback and kind words - we are very happy to see that our effort to create clear and informative figures is appreciated.
>
> **Re: Exploring invariance manifolds at lower levels of activation:**
>
> We agree that modifications of the objective function could enable exploration of broader questions about visual neurons coding properties. The objective you propose, incorporating a target (relative) activation level using the parameter $\lambda \in [0, 1]$ like $\left|\frac{\alpha_i}{\alpha_{\text{MEI}}} - \lambda\right|$, could indeed be used in our approach exactly as you suggested. Such an objective could allow us to investigate more extensively the complex response profiles of neurons, effectively exploring the level-sets of the high-dimensional response function. However, while this would be an interesting and insightful direction to explore, it presents some challenges, as invariances at $\lambda > 0$ can span large portions of the input space. For instance, at  $\lambda > 0$, images that both satisfy the norm constraint and achieve the target activation level could be generated in two ways: 1) by learning patterns only within the RF, which is the desired outcome, or 2) keeping the pattern within the RF similar to the case of $\lambda = 0$ but with a lower norm and “cheat” the norm constraint by introducing additional patterns outside the neuron’s RF. In the latter case, the features outside the RF represent uninformative contrast allocations unrelated to the neuron’s function and are merely artifacts of the procedure. Resolving this issue would require the development of additional mechanisms. Furthermore, at  $\lambda > 0$ , the invariance manifolds are likely to become very high-dimensional and exhibit a complex topology, significantly increasing the difficulty of capturing them effectively. Due to these challenges, exploring invariances at  $\lambda > 0$  extends beyond the scope of this work. Our focus on invariances at (or close to) the peak activation is motivated by three factors. First, this approach aligns with how invariances are often described in neuroscientific research. For instance, when one says that a complex cell present phase invariance, it is assumed that it is selective to Gabor stimuli with specific parameters and it is invariant to changes in phase of Gabor stimuli with specifically these parameters (i.e. the stimuli for it is particularly selective to). Second, investigating stimuli (and invariances) at peak activation is fundamental to understanding neural coding, as it reveals the features to which neurons are most selective and, therefore, offers critical insights into their functional role. Third, invariance manifolds at peak activation are likely to be lower dimensional, which makes their identification a more constrained yet meaningful problem, that enables us to focus on the alignment aspect of our approach for population-wide comparisons of single-neuron invariance manifolds.
>
> **Re: Related work using GANs:**
>
> Thank you for bringing these two works to our attention. Despite using a different approach, we agree that they are related to our work, as they also investigate neural invariances and response properties using generative models. We have included these works in our revised manuscript as part of the broader context of methods for studying neural invariances.
>
> **Re: Redundancy of the numerator of contrastive learning objective:**
>
> While we agree with your intuition that the INR's parametrization partially encourages smoothness, due to its nonlinear nature it can learn discontinuities (i.e. big changes in image space corresponding to small changes in the latent space). An objective function with only a diversity enforcing term cannot distinguish between solutions with and without discontinuities. This problem is resolved by the addition of an explicit term enforcing similarity between nearby images (the numerator) which discourages such discontinuities resulting in a smooth transformation along the learned invariance manifold.

---

> > ### Author Response · Authors · 2024-11-21
> > **Authors response to reviewer 5DFt (part 2)**
> >
> > **Re: Alignment of invariance manifolds vs MEIs:**
> >
> >  This is an important question, as it highlights the importance of using invariance manifolds rather than MEIs. In short, using MEIs alone can lead to grouping neurons together that should not be grouped. For example, consider a simple and a complex cell: the simple cell has a unique MEI, while the complex cell does not. It is possible that the optimized MEI of the complex cell happens to have the same phase as the unique MEI of the simple cell. Consequently, using MEIs (after alignment) results in a symmetric activation matrix, where the MEI of one neuron strongly activates the other. This symmetry causes these neurons to be grouped into the same cluster, even though they are functionally distinct. In contrast, the invariance manifold captures the entire set of stimuli that elicit peak activation. As a result, the corresponding activation matrix is asymmetric (as described in lines 268–272), and these cells will not be grouped into the same cluster. Importantly, this limitation of using MEIs is not confined to this specific example; it can occur with any pair of neurons that exhibit invariance properties and have only partially overlapping invariance manifolds (e.g., a complex cell and a rotation-invariant cell with a common gabor receptive field origin).
> >
> > **Re: How can this method be used beyond defining cell types?**
> >
> >  In general, our method allows for identifying meaningful shared directions in the input space that elicit a desired response across a group of neurons, accounting for specific spatial transformations of their RFs. While we have focused on affine transformations in this work, the formulation is flexible and can accommodate more complex transformations. The focus on clustering is one specific example of how the method can provide insights into the shared encoding properties of sensory neurons. Another potential application is comparing two groups of neurons with known labels (e.g., across species or brain areas) to quantify the extent to which their response profiles align. Another example is to study representational drift across days or task conditions by evaluating whether the invariance manifold of a neuron on day/task 1 aligns with its invariance manifold on day/task T. The key strength of our method, across these different applications, is that it will take invariances in the neural code into account and group/align neurons accordingly.
> >
> > **Re: Higher dimensional invariance manifolds:**
> >
> >  The dimensionality of the learned invariance manifold is a hyper-parameter that can be set to values greater than 1. Baroni et al. (2023) demonstrated that their method can recover 2D invariance manifolds (Figure 3 in their paper), highlighting the flexibility of this approach. Extending the approach to higher-dimensional invariance manifolds is an exciting direction for future research, as it could provide deeper insights into the full complexity of neural invariances while also advancing our ability to disentangle meaningful dimensions. However, this comes with challenges. For instance, in the current implementation it becomes computationally expensive to use the method for learning high-dimensional invariance manifolds. In this proof-of-concept study, we chose to focus on 1D invariances to prioritize visual clarity and interpretability, enabling us to effectively showcase and validate the method, specifically regarding the template matching and clustering aspects.

---

> > > ### Comment · Reviewer_5DFt · 2024-12-03
> > > **Thanks for the extensive response!**
> > >
> > > The reviewers appreciate the extensive response to our previous questions and the authors' willingness to incorporate our suggestions.
> > > Kudos on the beautiful paper and we will keep the score as such.

---

> > > > ### Author Response · Authors · 2024-12-03
> > > > **Thank you!**
> > > >
> > > > Thank you for your kind words and for taking the time to review our responses and updates. We appreciate your thoughtful feedback and positive evaluation of our paper.

---

### Official Review · Reviewer_oPYM · 2024-11-04

**Soundness:** 3
**Presentation:** 3
**Contribution:** 4
**Rating:** 8
**Confidence:** 5

**Summary:**

This paper adapts and builds on work that optimizes for an “invariance manifold” of maximally exciting inputs to neurons. The authors introduce an affine transformation that can be applied to learn a minimal similarity between one neuron’s invariance manifold and another’s. They show that this allows efficient clustering of neuron properties in simulations of macaque V1 responses and in neural predictions obtained from “digital twin” models of macaque V1 responses.

**Strengths:**

Overall, the idea behind the paper seems solid. It extends previous work of capturing invariances in an interesting direction of aligning the invariances between different neural populations. The paper is also fairly easy to read and understand, apart from the sections noted below.

**Weaknesses:**

As currently written, the paper muddles together previous work and what is new in this paper. For instance, in the abstract (lines 14, and followed up in line 21) it is noted that previous work does not model the invariances as a “continuous manifold”, and that this paper learns the continuous invariance manifold. However, I believe that the authors are directly using a method from Baroni et al 2023, which defines a continuous manifold so this isn't actually "novel" in this work. More details about weaknesses and ways to improve the paper are covered in the section section.

**Questions:**

A few easy-to-address concerns would move this paper into a clear acceptance for me:
1) The paper states that it tests the method on “biological neurons from macaque V1” (line 72). This is not true. The paper tests the method on *models* (i.e. additional simulations) trained to predict the response of V1 neurons (digital twins). There is no biological neural data in the paper to validate the method, and thus the findings may just be properties of the models and not properties of biological neurons. The wording in this section and elsewhere in the paper MUST be changed, as currently it is misleading to readers.
2)	Using the phrase “trivial” to describe known properties of receptive field attributes like position, size, and orientation does not seem appropriate. These coding properties are thought to be *fundamental* to the organization of neurons in primary visual cortex. For the second bullet point on line 56, Perhaps the authors mean something like “The method should allow for flexibility to account for known variability of neurons, and find common structure given this variability”?  Please rephrase this bullet point as you see appropriate, and also change other locations in the paper where these properties are referred to as “trivial”.
3)	I found the explanation of INRs (Lines 136-158) and the associated figure 1 very difficult to follow. This whole section could be improved to concretely explain what was done for this paper, instead of generally explaining INRs and differences from previous work (as the reader might not be familiar with the details of Baroni et al, so the methods should be self-contained in this paper). For instance, in Figure 1, the depiction of two neural networks is confusing, as I believe that the parameters of the response-predicting model $\theta$ are fixed, but this is not noted? Additionally, what images are used to train the INRs (is there a train/test set? I could not find details of this, or how these images were chosen for the two sets of experiments, in the main text or the supplement). It is also unclear how the modifications allow direct control of the spatial frequency (line 157), and if this is critical it should be better explained. These are some examples, but I would encourage the authors to generally revise this methods section and add appropriate details to the supplement so that the setup is clear to the readers.
4)	How are the parameters for the contrastive objective for “near” and “distant” points determined, and are different results obtained when using a different range for near and distant points? Additionally, it is stated that the formulation encourages a smooth and diverse learned manifold (218-219). It would be good if this was quantified, for instance by showing lines for real points like that in Figure 1 bottom right, to show that there are no jumps etc. when varying the latent parameters.
5)	In section 4.1, a few more details about the model neurons should be provided in the main text so that readers understand the simulations. Additionally, another sentence in the main text describing how the manifold alignment is measured (the right bars on the plots in Figure 2E,F & Fig 3 C) would be good to help  the reader understand the difference between the measured “Activation” and “Manifold” similarity.

I also had a few more general questions about the work:

6) How does this method hold up in the case of the underlying invariance manifold not being continuous in pixel space? For instance, take the simple example of $y=\texttt{min}(x^2, 1)$. Two points at +/-1 have the same y output (which is the maximum value) but they are not connected. In this sense, there are two discrete invariance manifolds of the maximum responses. I’m not sure how a  case like this would fit into the proposed framework, and so more detail about why a continuous manifold is a reasonable assumption (or at least acknowledging limitations of it) would strengthen the paper.
7) In terms of invariances, the work by Feather et al. (Neurips 2019, and Nature Neuro 2023) come to mind. However, this work found that deep neural networks often have idiosyncratic invariances that are not shared across different models (and they also did not attempt to characterize the full manifold of invariances—they just draw samples from it). Do the authors add in regularization priors (such as smoothness etc) to the initial MEI generation to promote more smooth and interpretable MEIs, which could encourage them to be more shared across the different models? Or is there something about the fact that the models used are an ensemble of CNNs, rather than a single one, discouraging idiosyncrasies? It also might be worth mentioning this previous work on invariances in the related work section.
8) Can some of the “affine transformations” for similarity of the manifolds be cast as a different similarity metric applied to the representations? The work by Williams et al. (NeurIPS 2021) on generalized shape metrics comes to mind. https://arxiv.org/pdf/2110.14739

Typos and other minor suggestions:
* Line 53:  “and identifying” is the wrong tense (does not match “compare”) – it should be “and identify”
* As currently written, the related work section is difficult to understand. It might be worth revisiting this section to unpack some of the jargon. Alternatively, some authors put the related work towards the end near the discussion, which in this case might help the reader understand why each of these points matters for the paper (for me, I had to return to this section after reading the rest of the paper to really understand what was going on).
* In Figure 1 bottom right panel, is this real data from one of the experiments or is it just a fake-data visualization? This should be noted.
* How is noise taken into account for the neural data?
* Lines 269-272: I believe wording in this section is too strong – the Gabor pattern of the simple cell will not always maximize the activity of a complex cell? Maybe rephrase, or more concretely explain what is meant.
* Line 429 – are there really “no invariances” in the Gabor patterns in Figure 3A? It seems a little strong to say this, as there are slight pixel changes.

---

> ### Author Response · Authors · 2024-11-21
> **Authors response to reviewer oPYM (part 1)**
>
> Thank you for taking the time to give us valuable feedback on many aspects of our submission. We are confident we can address all of your concerns, which we believe would make this a stronger submission and ultimately more understandable.
>
> ## Response to the main concerns:
>
> **Re: Novelty of the Continuous Invariance Manifold (Abstract Clarification):**
>
> You are right that learning a continuous invariance manifold has already been introduced by Baroni et al (2023), and in this paper we build on top of their approach to align and compare invariance manifolds across a population of neurons, uncovering shared invariance properties and allowing for clustering of neurons based on their invariances. It was not our intention to claim that method, and we have explicitly mentioned this in the main text multiple times (lines 64-66, 99, and 149). We agree that the abstract must be improved to make this point clear. We have therefore refined the abstract accordingly.
>
> **Re: The phrase “biological neurons from macaque V1”:**
>
>  You are, of course, right that we do not derive the properties from real neurons, but from a deep learning model trained on real neurons. We fixed the text to be more precise and make sure it is clear that we are applying our method on a response-predicting model of macaque V1 neurons. That said, the type of response-predicting model we use has been used in other works in monkey [1,2] and other species [3,4,5] where its results have been verified with biological neurons, including specifically that MEIs the model generates activate highly corresponding neurons in closed loop experiments. In addition, some of the results in our paper (i.e. simple and complex cell clusters) are well known. Thus, while we agree that any biological statement can only derive from experimentally tested results, these models have proven to be quite accurate in the past.
>
> [1] Willeke, Konstantin F., et al. "Deep learning-driven characterization of single cell tuning in primate visual area V4 unveils topological organization." bioRxiv (2023): 2023-05.
>
> [2] Bashivan, Pouya, Kohitij Kar, and James J. DiCarlo. "Neural population control via deep image synthesis." Science 364.6439 (2019): eaav9436.
>
> [3] Walker, Edgar Y., et al. "Inception loops discover what excites neurons most using deep predictive models." Nature neuroscience 22.12 (2019): 2060-2065.
>
> [4] Franke, Katrin, et al. "State-dependent pupil dilation rapidly shifts visual feature selectivity." Nature 610.7930 (2022): 128-134.
>
> [5] Fu, Jiakun, et al. "Pattern completion and disruption characterize contextual modulation in the visual cortex." bioRxiv (2023): 2023-03.
>
> **Re: Using “trivial” to refer to known properties of RF attributes:**
>
> We completely agree that these are fundamental properties of receptive fields. What we meant by the word ‘trivial’ was that they are already very well known – in the sense of “obvious”. Therefore, we want to study additional less understood characteristics of sensory coding (invariance) properties across neurons irrespective of these well known properties. To avoid that the use of the word “trivial” might be read as a condescending statement (which we do not intend), we changed “trivial” to “nuisance” to better express what we intend to say, and accordingly rephrased the second bullet point.
>
>
> **Re: Improving the method section and missing information in Figure 1:**
>
> The parameters of the response-predicting model $\theta$ are indeed fixed, which is now explicitly noted in the revised version. The INR is not trained on any dataset of images; instead, it serves as a generative model that maps pixel coordinates to pixel values (e.g., grayscale intensities), producing images as continuous functions of pixel coordinates (Lines 158–161, 203-207). In this work, following the approach of Baroni et al. (2023), the INR is trained to generate diverse images along the invariance manifold of a neuron. We have clarified the purpose of using the INR early in the method section so that it is easier to understand for readers unfamiliar with Baroni et al (2023). Additionally, the use of random Fourier projection on pixel coordinates $(x,y)$ to control spatial frequency, as introduced by Tancik et al. (2020) and employed by Baroni et al. (2023), supports accurate representation of the invariance manifold. While not the primary focus of this paper, we adopted a setup inspired by Baroni et al. (2023), with some refinements, to capture the invariance manifold of a single neuron effectively. We have refined the revised manuscript to ensure the method section is self-contained, accessible, and highlights these refinements clearly.

---

> > ### Author Response · Authors · 2024-11-21
> > **Authors response to reviewer oPYM (part 2)**
> >
> > **Re: Different configurations of “near” and “distant” points in the contrastive objective and the effect on the smoothness of the learned invariance manifold:**
> >
> > While the configuration we used for specifying “near” and “distant” points in the contrastive objective yields robust results—promoting diversity in the learned manifold such that the complete underlying invariance manifold is captured, as shown in Figure 2E—we agree that quantifying how this configuration affects the learned invariance manifold would add valuable insight. This is particularly relevant given similar feedback from reviewers 5DFt and Nobq. To address this, we have performed an analysis (Appendix G) to evaluate how different configurations of “near” and “distant” points influence the ability of our method to capture the underlying ground truth manifold.
> >
> >
> > **More details about the model neurons and manifold alignment in the main text:**
> >
> > To address your concern and help the reader better understand both the nature of the simulated neurons as well the manifold alignment metric, minimizing the need to refer to the appendix, we have included a brief description of the simulated neurons (e.g., modeled invariances) and explicitly explained how both the activation and the manifold similarity metrics are computed.
> >
> > &nbsp;
> >
> > ## Response to the more general questions:
> >
> > **Re: Capturing non-continuous invariance manifolds:**
> >
> > This is a great point and one that has already been investigated and addressed by Baroni et al. (Appendix E). Specifically, they show that even though the method is designed to capture a continuous manifold, it can successfully capture (an approximation of) an underlying discrete manifold, by introducing/learning sudden jumps. This happens by shifting the emphasis from the contrastive objective as the INR is trained. Specifically, the emphasis on the contrastive objective at the beginning of the training encourages diversity while at the same time keeping transitions smooth, potentially at the expense of activation. For a neuron with a non-continuous manifold, activations in this stage of the training will not be high, on average. As training advances, the importance of activation objective becomes more prominent and the INR learns to introduce "jumps” to capture (an approximation of) the ground truth non-continuous manifold to satisfy the activation objective (i.e. achieve maximal activation averaged across sampled images along the learned invariant manifold). This is implemented by a scheduler which we describe in Appendix B.
> > &nbsp;
> >
> > **Re: Idiosyncrasies of MEI and invariances:**
> >
> > The MEIs are obtained by directly optimizing pixel values, for which we use an ensemble ($n=3$) of response-predicting models to discourage idiosyncrasies, with no additional smoothness regularizations. For learning the invariance manifold, we use an INR as a continuous function over pixel coordinates (i.e., it maps pixel coordinates to grayscale values), which can inherently encourage smoothness since nearby pixels (i.e., similar input values) yield similar outputs due to the continuity of the MLP, serving as an additional factor that discourages idiosyncrasies. Finally, we appreciate your suggestion to discuss the works by Feather et al. (NeurIPS 2019, Nature Neuro 2023). We believe the two approaches can be complementary: our method provides a tool for investigating computational principles underlying invariances in biological networks, which could potentially inform the design of artificial neural networks and help reduce the discrepancies between biological and artificial systems, as highlighted by Feather et al.
> > &nbsp;
> >
> > **Re: The approach of Williams et al. (2021) vs ours:**
> >
> > While their framework and ours both involve representational analysis, they answer distinct questions: Williams et al. cluster **entire networks or brain regions** by measuring overall representational similarity across inputs, using a metric space framework. Their goal is a broad, systematic comparison across diverse networks or brain areas, relying on metrics that satisfy properties like the triangle inequality. ​​In contrast, our work clusters **individual neurons within a specific population** based on **activation-driven functional alignment** of their invariance manifolds, prioritizing activation maximization rather than structural similarity. Our objective is to identify shared functional properties within a neural population, rather than structural similarity across diverse systems. In sum, Williams et al. address representational similarity across large, heterogeneous systems, while our work focuses on functional alignment within a specific neural population, making the two approaches complementary but not directly comparable.

---

> > > ### Author Response · Authors · 2024-11-21
> > > **Authors response to reviewer oPYM (part 3)**
> > >
> > > ## Response to the minor suggestions/questions:
> > >
> > > - **Re: Typos**: Thank you for pointing this out. Other reviewers have also noted some typos and we will fix all of them in the revised version of our manuscript.
> > > - **Re: Improving the related work section:** Thank you for the suggestion. In the revised manuscript, we tried to improve the related work section such that it helps the reader to better understand (the contribution of) our method in the context of related past work.
> > > - **Re: Is Figure 1 bottom right panel real data or just an illustration?** This figure is just an illustration (albeit it is similar to the actual results, as it would be expected). We do refer to this figure as a "schematic representation of the neurons’ responses" in the caption of Figure 1.
> > > - **Re: How is noise taken into account for the neural data?** The response-predicting model we use is trained to maximize a Poisson conditional likelihood objective $p(r|x)$ where $x$ is the stimulus (i.e. natural images) shown to the macaque and $r$ is the corresponding recorded population response. Therefore, as we use a trained response-predicting model, the invariance manifolds are learned on estimated conditional firing rates (i.e. noiseless responses) rather than noisy responses.
> > > - **Re: The wording regarding simple vs complex (Gabor) cells:** What we meant to say here is that if we consider a simple cell and a complex cell **after the alignment** of their invariance manifolds—meaning their nuisance RF attributes (i.e., position, orientation, and scale) are “matched”—then the invariance manifold of the simple cell (essentially a single image which is its MEI) will maximally activate the complex cell, as it represents a single point along the invariance manifold of the complex cell. This argument follows directly from the previous sentence and paragraph, which discuss the activation matrix obtained after aligning the invariance manifolds. We have improved the text to make this example more clear.
> > > - **Re: ”no invariances” in the Gabor patterns:** You are right that some slight variations can be observed for the simple cells, which can happen as we are requiring the generated images to be near-maximally exciting. However, the variations are so small that it is virtually equivalent to having no invariance.

---

> > > > ### Comment · Reviewer_oPYM · 2024-11-26
> > > > **response to authors rebuttal**
> > > >
> > > > Thank you to the authors for making the wording more clear and precise in the paper. I believe your changes strengthen the paper and will make it easier for readers to understand the methods and results.
> > > >
> > > > The analysis in Appendix G is quite interesting, and the validations in F of that figure might be nice to include in the main text for Figures 2 and/or 3 if there is space in the camera ready version. I found the sharp change in the stimuli for 0.40 and 0.45 quite interesting. Do you have a sense of why this is occuring? I guess I would have expected all stimuli to look the same, rather than sharp changes (this is not critical to explain, I'm just curious).
> > > >
> > > > Re: My question mentioning WIlliams et al. I was thinking more about the "metric" used and less about the application, but maybe it wasn't quite the right reference. For instance, the Procrustes distance (https://en.wikipedia.org/wiki/Procrustes_analysis) will remove translational, rotational, and scaling components to perform the alignment. This seems interesting to think about, but as you said, it might be complementary but not explicitly related.
> > > >
> > > > In general, congrats on a very interesting paper! I have updated my scores appropriately.

---

> ### Author Response · Authors · 2024-11-27
> **Thank you!**
>
> Thank you for your response, your kind words, and for raising your score!
>
> **Re: sharp change in the stimuli for 0.40 and 0.45:** This is an interesting point. Our intuition is that while having all images be the same will maximize the numerator of the contrastive objective, it results in the worst possible denominator. In contrast, a solution introducing jumps at periodicity equal to half of the latent space (similar to the figure) ensures that the denominator is optimal (since for every $z_i$ its “distant” point is exactly located at half latent space distance and therefore has opposite polarity), but does not result in the optimal numerator.
> Then the question is: why is the second solution (i.e., having jumps) preferred over the first one (i.e., no jump with all images the same)? Let us consider the argument inside the $\log$ of the contrastive learning objective, which in the above mentioned cases is simple to compute:
> - **First solution where all images look the same:** Both numerator and denominator are $\exp(1/\tau)$, resulting in a final value of $1$.
> - **Second solution with jumps:** Half of the “near” images will have a cosine similarity of $1$ and half of them $-1$, resulting in a numerator of $\left(\exp(1/\tau) + \exp(-1/\tau)\right)/2$, and a denominator of $\exp(-1/\tau)$, resulting in $\left(\exp(2/\tau) + 1\right)/2 > 1$ for temperature $\tau \gt 0$.
>
> Therefore, in terms of optimizing the contrastive objective the second solution is preferrable.
>
> **Re: "metric" used for measuring similarity by Williams et al. and Procrustes analysis:** At a high level, we believe, these methods can be viewed as optimizing transformations to align representations with the goal of evaluating a distance metric: $\min_{T_1, T_2} D(T_1(x_1), T_2(x_2))$, where $x_1$ and $x_2$ represent shapes, representations, or (in our case) invariance manifolds. Procrustes analysis constrains $T$ to translation, rotation, and scaling with $D$ as a geometric distance in Euclidean space. Williams et al. generalize this framework by defining transformations  $T_1$  and  $T_2$  that map neural representations into a shared feature space, ensuring that the resulting metrics satisfy properties like symmetry, equivalence, and the triangle inequality, thereby enabling robust comparisons between representations. Our template alignment step fits into this broader view but operates in functional activation spaces, where $D$ is a **neuron-specific metric** embedded in the functional (activation) space, and we optimize for: $\min_{T_{2,1}} D(F_1(x_1), F_1(T_{2,1}(x_2))) \equiv \min_{T_{2,1}} D_1(x_1, T_{2,1}(x_2)) \equiv \max_{T_{2,1}} F_1(T_{2,1}(x_2))$
> where $F_1$ is the response-predicting model of neuron $1$ and $T_{2,1}$ is an affine transformation acting on the invariance manifold of neuron $2$ with the objective of maximizing the response $F_1(T_{2,1}(x_2))$ of neuron 1 (or equivalently minimizing its distance to the response $F_1(x_1)$ of neuron $1$ to its own maximally exciting manifold $x_1$). Overall, in contrast to the other two approaches, which align representations or shapes to evaluate their similarity, our approach aligns single-neuron invariance manifolds to gain insights into functional similarities across neurons, rather than measuring similarities between the manifolds themselves.
>
> Thanks again for your thoughtful comments!

---

### Author Response · Authors · 2024-11-21
**Authors general response (part 2)**

**Regarding 4:** We improved the methods section to ensure it is self-contained and accessible to readers unfamiliar with prior work that our method is based on. Among many changes to improve clarity, we clarified that the parameters of the response-predicting model are fixed, and explicitly stated that the INR is not trained on any dataset of images but is trained to generate the images that maximally activate the modeled neuron. Furthermore, we improved the description of how random Fourier projection is employed to control spatial frequency in the INR.
In the results section, we added a concise description of the implementation of simulated neurons and explicitly described both the activation and the manifold similarity metric. These changes improve readability and reduce reliance on the appendix

**Regarding 5:** To address concerns about the contrastive regularization objective and its role in learning smooth invariance manifolds, specifically with regards to the definition of “near” and “distant” points, we conducted an analysis (Appendix G) to evaluate how different definitions influence the ability of our method to capture the underlying invariance manifold. This analysis demonstrates the robustness of our approach to different choices of these definitions while highlighting their impact on capturing the complete manifold.

**Regarding 6:** Reviewer **aC4g** proposed a structured comparison across different levels of affine transformation complexity to explicitly demonstrate where our method outperforms Klindt et al. (2017) and Ustyuzhaninov et al. (2019). While it is clear from the design of our method (and results in Figure 2) that it is inherently more flexible than these past methods, this analysis serves to further empirically verify what we theoretically understand about their limitations. Additionally, it will highlight one of the key advantages of our approach—its model-agnostic nature. Unlike the previous methods, which rely on specific model architectures and weight vectors, our method directly uses neural activations to align and cluster neurons. This not only allows it to work across diverse response-predicting models but even enables it to outperform the original methods when applied to their own architectures. We are currently working on this comparison and plan to update the manuscript accordingly. While time constraints may prevent us from completing the analysis within the limited rebuttal period, we are committed to performing this study and will communicate our results to the reviewers as soon as possible.

We hope that our responses address all potential concerns. Please let us know if you need any further clarifications.

---

> ### Author Response · Authors · 2024-11-24
> **Follow up regarding point 6 (comparison of our method to model-based clustering methods)**
>
> We are happy to let you know that we managed to perform the analysis that was requested by reviewer **aC4g** (point 6 above), comparing our method to that of Klindt et al. (2017) and Ustyuzhaninov et al. (2019), and have included it in the new revision of our manuscript (Appendix H). The results empirically demonstrate the flexibility of our approach in accounting for affine transformations (similar to the results in Figure 2 of the initial submission), while highlighting that the other two methods can only account for shifts (Klindt et al. 2017) and rotations (Ustyuzhaninov et al. 2019). Importantly, we applied our approach on both architectures from Klindt et al. (2017) and Ustyuzhaninov et al. (2019), and show that on both models our method yields consistent clusters, highlighting the model-agnostic nature of our approach.
>
> Finally, we would like to emphasize that accounting for the full spectrum of affine transformations is not the only advantage of our method over Klindt et al. (2017) and Ustyuzhaninov et al. (2019). Unlike our method, the previous methods are model-based (i.e., not model-agnostic), do not explicitly disentangle invariances from other computational properties (e.g., tuning), and can incorrectly group similar neurons into different clusters due to potential redundancies in the features learned by the model. Please refer to our previous response to reviewer **aC4g** for more details.
>
> With this analysis, we believe, all issues raised by all the reviewers are addressed. We would like to thank all the reviewers again for their thoughtful feedback and valuable suggestions. Please let us know if you need any further clarification.

---

### Author Response · Authors · 2024-11-21
**Authors general response (part 1)**

We would like to thank all the reviewers for their thoughtful and detailed feedback on our submission. We were pleased to see that the reviewers found our manuscript to be “well-written” (Rev **aC4g**) and “easy to read” (Rev **oPYM**), with figures that are “stunning and visually pleasing” (Rev **5DFt**), “very clean and show convincing results” (Rev **aC4g**). Additionally, we were happy that our method was recognized as “innovative”, “thoroughly evaluated” (Rev **5DFt**) and “solid” (Rev **oPYM**), providing an “elegant” (Rev **5DFt**) systematic approach to study neuronal invariances at the population-level (Rev **5DFt** and **Nobq**), with the potential to characterize new cell types (Rev **Nobq** and **aC4g**).

The reviewers also raised several valid issues. The main points were:

1. Refinements in terminology to avoid misinterpretations and improve rigor (Reviewer **oPYM**).
2. A clearer introduction of the term “invariance manifold” and highlighting the fact that the concept of invariances extend beyond invariances at peak activation (Reviewers **5DFt**, **Nobq**, and **aC4g**).
3. Improving the related work section by adding references to additional studies and enhancing clarity in describing limitations of prior methods (Reviewers **5DFt** and**aC4g**).
4. A more comprehensive explanation of the methodology to ensure the methods section is self-contained and clear (Reviewers **oPYM** and **Nobq**).
5. Clarification on the contrastive regularization objective, including its importance for learning smooth invariance manifolds and the robustness of results to different configurations of “near” and “distant” points (Reviewers **oPYM**, **5DFt**, and **aC4g**).
6. Performing a specific comparison of our method with two past model-based clustering methods (Reviewer **aC4g**)

With the exception of point 6 (discussed in detail below, and for which we are currently working on and have already communicated our plan to Rev **aC4g**), we have addressed all the concerns raised by the reviewers which we believe have improved the quality of our submission. While we provide a summary here, please also refer to our more detailed responses to each individual review.

**Regarding 1:** To avoid any potential misinterpretation, in the revised manuscript we made sure to more strongly emphasize that our results are obtained by applying our method on a response-predicting model of macaque V1 neurons (digital twin), and are not obtained directly from biological neurons. In our response to reviewer **oPYM**, we included references to previous studies in which similar analysis performed of digital twins resulted in findings that generalize to their biological counterparts (as validated by in in-vivo experiments). To address concerns about the term “trivial” in relation to canonical RF attributes in V1 (e.g. position, size, and orientation), we replaced it with “nuisance,” which better acknowledges the foundational importance of these well-known properties while highlighting that their variability across neurons poses a challenge to identifying similarities in invariance properties. These changes aim to enhance both clarity and precision in the manuscript.

**Regarding 2:** We clarified the concept of “invariance manifold” by explicitly defining the term in the revised manuscript prior to its first usage and including a concrete example to improve clarity for readers unfamiliar with the concept. In our response to the reviewers, we provided a detailed explanation of the challenges associated with exploring invariances at suboptimal response levels and justified our focus on invariances around maximally exciting stimuli. Additionally, in the main text, we acknowledged that invariances can also be studied at other response levels, which is however outside of the scope of this study, and more concisely justified our decision to focus on maximal response levels.

**Regarding 3:** We introduced several modifications to the related work section to better position our proposed method within the context of prior works, highlighting both its similarities and differences with other approaches while emphasizing our contributions. As suggested by Rev 5DFt, we incorporated references to two additional studies that use GANs to investigate neural invariances and their geometry. Additionally, in our response to Rev **aC4g**, we elaborated on the differences between our method and that of Ustyuzhaninov et al. (2019) in achieving functional clustering based on neural response properties.

---

### Meta-Review · Area_Chair_UQ26 · 2024-12-21

**Metareview:**

This paper describes a novel method for identifying single-neuron invariances and functional clusters, with application to recordings from macaque V1.  All four reviewers were enthusiastic about the the paper, and I'm pleased to report that it has been accepted to ICLR.  Congratulations!  Please revise the manuscript to address all reviewer comments and discussion points where possible.

**Additional Comments On Reviewer Discussion:**

Reviewers were unified in their appreciation for the paper. Comments focused mainly on clarifications of the relationship to previous work and suggestions for new experiments to demonstrate the utility of the proposed method. The authors did a very thorough job of addressing reviewer comments.

---

### Decision · Program_Chairs · 2025-01-22

Accept (Oral)